# Mrg15 stimulates Ash1 H3K36 methyltransferase activity and facilitates Ash1 Trithorax group protein function in *Drosophila*

Chang Huang[1], Fu Yang[2], Zhuqiang Zhang[1], Jing Zhang[1], Gaihong Cai[2], Lin Li[2], Yong Zheng[1], She Chen[2], Rongwen Xi[2] & Bing Zhu[1,3]

Ash1 is a Trithorax group protein that possesses H3K36-specific histone methyltransferase activity, which antagonizes Polycomb silencing. Here we report the identification of two Ash1 complex subunits, Mrg15 and Nurf55. In vitro, Mrg15 stimulates the enzymatic activity of Ash1. In vivo, Mrg15 is recruited by Ash1 to their common targets, and Mrg15 reinforces Ash1 chromatin association and facilitates the proper deposition of H3K36me2. To dissect the functional role of Mrg15 in the context of the Ash1 complex, we identify an Ash1 point mutation (Ash1-R1288A) that displays a greatly attenuated interaction with Mrg15. Knock-in flies bearing this mutation display multiple homeotic transformation phenotypes, and these phenotypes are partially rescued by overexpressing the Mrg15-Nurf55 fusion protein, which stabilizes the association of Mrg15 with Ash1. In summary, Mrg15 is a subunit of the Ash1 complex, a stimulator of Ash1 enzymatic activity and a critical regulator of the TrxG protein function of Ash1 in *Drosophila*.

[1] National Laboratory of Biomacromolecules, CAS Center for Excellence in Biomacromolecules, Institute of Biophysics, Chinese Academy of Sciences, Beijing 100101, China. [2] National institute of Biological Sciences, Beijing 102206, China. [3] College of Life Sciences, University of Chinese Academy of Sciences, Beijing 100049, China. Chang Huang, Fu Yang and Zhuqiang Zhang contributed equally to this work. Correspondence and requests for materials should be addressed to R.X. (email: xirongwen@nibs.ac.cn) or to B.Z. (email: zhubing@ibp.ac.cn)

I n *Drosophila*, homeotic (Hox) proteins, which are homeodomain-containing transcription factors, are master regulators of body segment specification. The spatially restricted expression pattern of Hox genes along the anterior-posterior axis of the organism, which are determined by upstream transiently expressed segmentation factors during early embryogenesis, can be maintained throughout the subsequent developmental stages. Both the loss and ectopic expression of Hox genes leads to cell fate changes and homeotic transformation[1,2]. Polycomb group (PcG) and Trithorax group (TrxG) proteins are two groups of factors with opposing functions and maintain the transcriptional "off" and "on" states of Hox genes, respectively[1–4]. Genetic studies have revealed that the PcG and TrxG proteins function through intra-group cooperation and inter-group antagonism to maintain the transcriptional balance of Hox genes, and biochemical studies have revealed that many PcG and TrxG proteins are organized into multi-subunit complexes with diverse chromatin regulating capacities, such as histone-modifying and ATP-dependent chromatin remodeling activities[1–5]. In addition to the above-mentioned biochemical characterization of these PcG and TrxG protein complexes, the crosstalk and autoregulation of these complexes are highly interesting[6–13].

The absent, small, and homeotic-1 (*ash1*) gene was first identified as a locus in *Drosophila* at which mutations caused small-disc, discless and homeotic transformation phenotypes similar to mutations at the *trithorax* locus[14,15]. Ash1 was classified as a TrxG protein based on three criteria: (1) the homozygote of the *ash1* weak allele demonstrated homeotic transformation phenotypes; (2) the trans-heterozygous mutant of *ash1* and *trithorax* alleles demonstrated homeotic transformation; and (3) the heterozygous *ash1* strong allele suppressed the phenotype of *polycomb* null flies[16]. The *ash1* gene encodes a SET domain-containing protein[17], which was later confirmed as a nucleosomal histone methyltransferase that catalyzes H3K36me2[12,18–21]. Ash1 is vital to counteract PcG proteins[20,22,23], and the derepression function of Ash1 depends on histone methyltransferase activity, because flies with mutations at the catalytic center of Ash1 display homeotic transformation phenotypes[17] and H3K36me2 catalyzed by Ash1 directly inhibits PRC2 catalyzed H3K27me3 establishment[11,12]. Although several genetic and physical interaction partners have been reported[20,24–27], whether Ash1 exists in a stable protein complex, similar to many other PcG and TrxG proteins, remains unclear.

Here we report the purification and characterization of an Ash1-containing protein complex from *Drosophila* S2 cells. We identify Mrg15 and Nurf55 as integral components of the Ash1 complex, and Mrg15 activates Ash1. The composition of the Ash1 complex and the role of Mrg15 in the activation of the histone methyltransferase activity of Ash1 are conserved in mammals. Moreover, this activation mechanism, which is mediated by Mrg15, is required for histone methyltransferase activity and TrxG protein function of Ash1 in vivo.

## Results

### *Drosophila* Ash1 forms a stable complex with Mrg15 and Nurf55.
*Drosophila* Ash1 is a large protein comprising 2226 amino acids. We first generated a *Drosophila* S2-derived cell line stably expressing a Flag-tagged truncated form of Ash1 (Ash1C, amino acids 1146–2226) protein (Fig. 1a) to explore the possibility that Ash1 may function within a protein complex in vivo. Then, we performed affinity purification with antibodies against Flag under high stringency (extensive washing with 500 mM KCl). The silver staining results demonstrated a major protein band of ~55 kDa that was co-purified with Flag-Ash1C (Fig. 1b).

Mass spectrometry analysis of this band identified peptides from two proteins: Mrg15, a chromo domain and MRG domain-containing protein, and Nurf55, a WD40 domain-containing protein (Fig. 1b). We generated antibodies against *Drosophila* Ash1 and Mrg15 and the presence of Nurf55 and Mrg15 in the Ash1 complex was validated by western blotting (Fig. 1c). We also generated a Flag-tagged full-length Ash1 stable S2 cell line (Supplementary Fig. 1a) and performed affinity purification. Again, Nurf55 and Mrg15 were co-purified (Supplementary Fig. 1b), but no additional stable subunits were subsequently confirmed, which suggested that Ash1, Mrg15, and Nurf55 constitute the core Ash1 complex. The complex formation was further confirmed by Co-Immunoprecipitation (Co-IP) of the endogenous proteins (Fig. 1d).

### Ash1 complex is conserved in mammals.
The human Ash1 homolog is termed hAsh1L (2964 amino acids), and we failed to express the full-length protein in HEK293 cells. Nevertheless, we were able to express the Flag-tagged C-terminal region of hAsh1L (amino acids 1465–2964, Flag-hAsh1L-C), which shares a high sequence similarity with *Drosophila* Ash1C and contains all known domains of the Ash1 protein. Affinity purification and subsequent mass spectrometry analysis identified four co-purified proteins (Supplementary Fig. 1c). Two of them were both MRG domain-containing proteins: MORF4L1 and MORF4L2 (also named as hMrg15 and hMrgX, respectively), human homologs of *Drosophila* Mrg15. Notably, hMrg15, but not hMrgX, contained an N-terminal chromo domain. The other two subunits were both WD40 domain-containing proteins: RbAp46 and RbAp48 (also known as RBBP7 and RBBP4, respectively), human homologs of Nurf55. These results were further confirmed by Co-IP experiments (Supplementary Fig. 1d). Co-IP experiments revealed that the hMrg15 and hMrgX proteins did not interact with each other (Supplementary Fig. 1e), which suggests that they both could form complexes with hAsh1L in a mutually exclusive manner. On the other hand, RbAp46 and RbAp48 often coexist in many chromatin-modifying complexes and histone chaperone complexes and they interact with histone H4 and facilitate the binding of chromatin regulators to the corresponding substrates[28–30]. Taken together, the subunit composition of Ash1 complex is conserved in mammals, suggesting an evolutionarily conserved function.

Mrg15 (also known as Eaf3 in yeast) contains an N-terminal chromo domain that has been reported to recognize H3K36me2/3, although the binding affinities of Mrg15 for methylated H3K36me2/3 peptides are fairly low ($K_d \approx 200$–$400\,\mu$M)[31–33]. Mrg15 also contains a C-terminal MRG domain that is mainly responsible for protein–protein interactions[34–37]. Mrg15 and the yeast homolog Eaf3 have been reported to be subunits of two other histone-modifying enzyme complexes: the NuA4 histone acetyltransferase complex[38–41] and the Rpd3S (or Sin3a in higher eukaryotes) histone deacetylase complex[42–46].

### Mrg15 activates histone methyltransferase activity of Ash1.
To understand the biochemical role of these subunits in the context of the Ash1 complex, we purified recombinant *Drosophila* Ash1C (1146–2226 aa), Mrg15 and Nurf55 proteins and performed histone methyltransferase activity assays using recombinant oligonucleosomes (RON) as substrates. Interestingly, the addition of Mrg15, but not Nurf55, dramatically stimulated the enzymatic activity of Ash1C (Fig. 2a). In a control experiment, Mrg15 displayed no histone methyltransferase activity (Supplementary Fig. 2a), in consistence with its lack of SET domain. Moreover, the coexpressed and co-purified Ash1C/Mrg15 complex displayed much higher activity than Ash1C alone (Fig. 2b). To examine

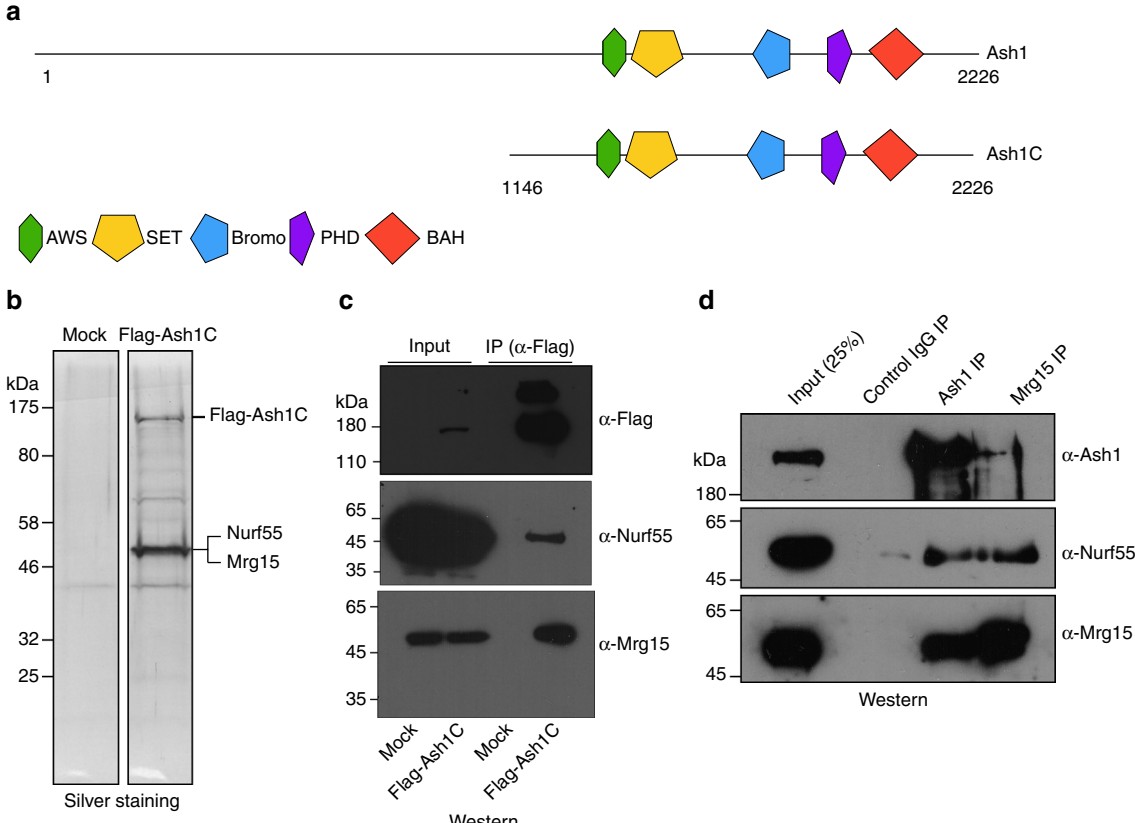

**Fig. 1** Ash1 forms a stable complex with Mrg15 and Nurf55. **a** Schematic map shows the protein domain organization of *Drosophila* Ash1. **b** Silver staining results of affinity purified materials from S2 cells expressing mock- or Flag-Ash1C. **c** Western blot results using antibodies against Flag, Nurf55 and Mrg15 to confirm the mass spectrometry results from affinity purification. **d** Endogenous Co-IP results with antibodies against Ash1 and Mrg15 using S2 cell nuclear extracts as input

whether Mrg15 changes the product specificity of Ash1C, we analyzed the reaction products with antibodies specific to each state of H3K36 methylation. Ash1C/Mrg15 complex displayed an increased activity in producing H3K36me1 and H3K36me2, but remained to be incapable of producing H3K36me3 (Supplementary Fig. 2b), similar to the activity of Ash1 alone that was previously characterized[12,18,21].

Given that the subunit composition of Ash1 complex is highly conserved between *Drosophila* and human, we evaluated whether hMrg15 and hMrgX proteins could stimulate Ash1 activity in vitro. Both recombinant hMrg15 and hMrgX stimulated the enzymatic activity of *Drosophila* Ash1C (Supplementary Fig. 2c). These results indicated that not only the complex composition, but also the mechanism of enzymatic activity regulation was evolutionarily conserved. Notably, the hMrgX protein natively lacking a chromo domain stimulated the activity of Ash1C, which suggested that Mrg15 without the chromo domain could stimulate Ash1C. Indeed, the recombinant *Drosophila* Mrg15 mutant with the chromo domain deleted (Mrg15Δchromo) was capable of activating Ash1C (Fig. 2c), albeit in a less efficient manner (Supplementary Fig. 2d, e). Notably, Mrg15 protein lacking MRG domain (Mrg15ΔMRG) completely lost the stimulation activity (Supplementary Fig. 2d). These results suggest that the MRG domain of Mrg15 is the primary activator of Ash1, while the chromo domain of Mrg15 may participate in the stimulation process by stabilizing the structure of the Ash1-Mrg15 enzyme complex.

We also measured the reaction kinetics of the Ash1C and Ash1C-Mrg15 complex by titrating substrates (Fig. 2d, Supplementary Fig. 2f). The incorporation of Mrg15 robustly increased the reaction $V_{max}$ and the difference was statistically significant ($P = 0.018$, Student's $t$-test), but it had a much milder effect on the $K_m$ and the difference was not statistically significant ($P = 0.79$, Student's $t$-test) (Fig. 2e).

**Ash1 recruits Mrg15 to their common target genes**. To test the relationship between Mrg15 and Ash1 in vivo, we performed ChIP-seq experiments using wild type and Ash1 knockdown and Mrg15 knockdown S2 cells (Supplementary Fig. 3a, b, Supplementary Tables 1, 2). In total, we identified 3678 Mrg15 peaks, which was far greater than the number of Ash1 peaks (562) in the genome (Fig. 3a). Moreover, we plotted Ash1 intensity at all Mrg15 peaks and no apparent enrichment of Ash1 was observed (Supplementary Fig. 3c). The above results likely reflect the facts that Mrg15 exists in at least three different chromatin-modifying complexes (NuA4 histone acetyltransferase complex, Sin3a histone deacetylase complex and Ash1 histone methyltransferase complex) and Ash1 complex only contributes to a small percentage of Mrg15 chromatin localization.

In contrast, 41% of the Ash1 peaks overlapped with Mrg15 peaks (Fig. 3a). We also plotted Mrg15 intensity at all Ash1 peaks, and Mrg15 signal was clearly enriched in the majority of Ash1 peaks, even at the weak Ash1 peaks (Fig. 3b, c, Supplementary Fig. 3d). These results suggested that Mrg15 was present even at the majority of the 59% of Ash1 peaks that were not initially called as Mrg15 peaks, likely due to arbitrary cutoff used during peak calling. Indeed, at all Ash1 peaks, Mrg15, and Ash1 intensity

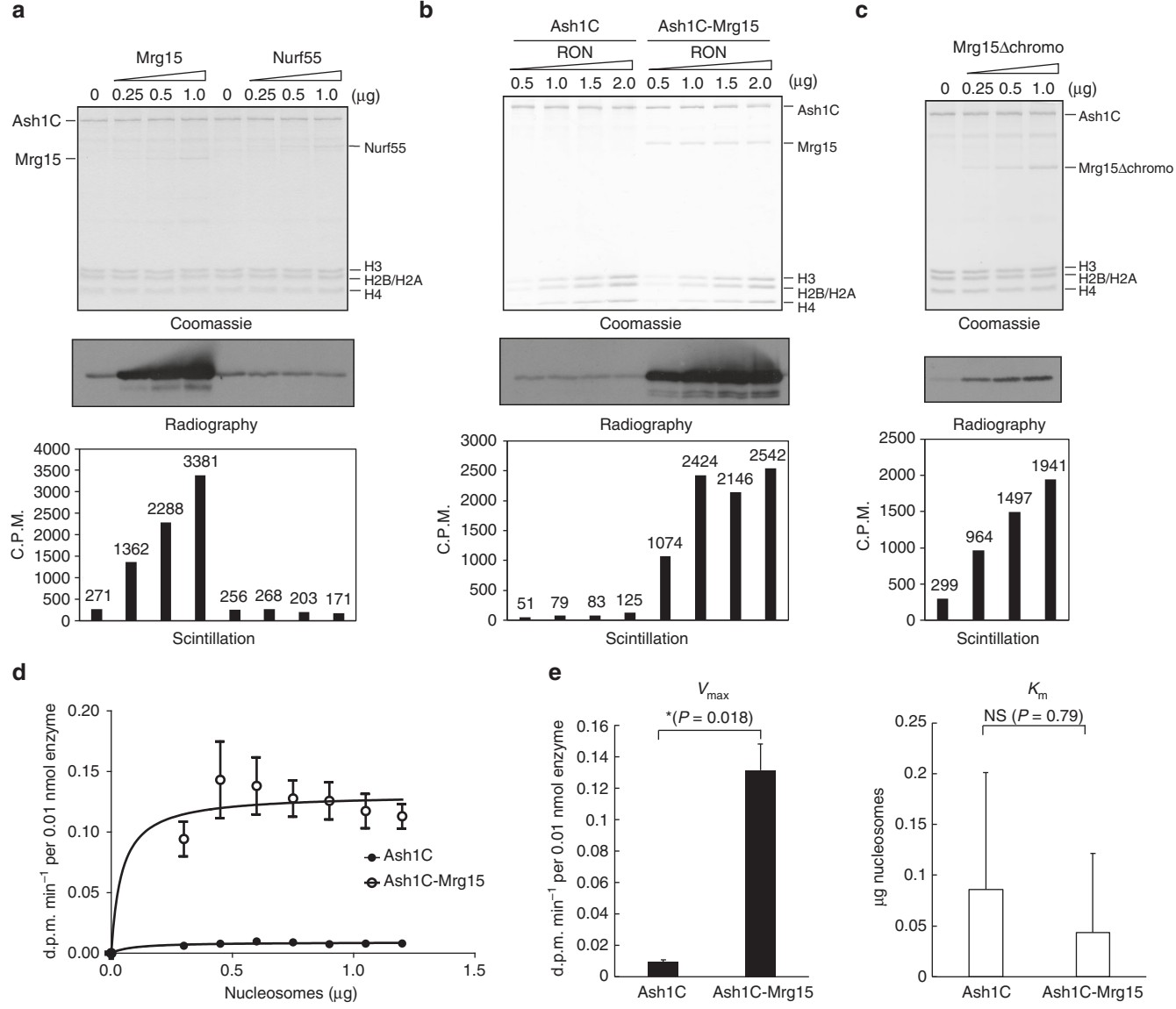

**Fig. 2** Mrg15 stimulates the histone methyltransferase activity of Ash1C in vitro. **a** Radiography and scintillation results show that recombinant Mrg15, but not Nurf55, stimulates the activity of Ash1C. **b** Radiography and scintillation results show that Ash1C-Mrg15 complex has stronger enzymatic activity than Ash1C alone. **c** Radiography and scintillation results show Mrg15Δchromo can stimulate the activity of Flag-Ash1C-His. **d** Kinetic characterization of Ash1C and Ash1C-Mrg15 complex with titration of recombinant oligonucleosomes. Data are plotted as mean ± s.e.m. ($n = 2$). **e** $V_{max}$ and $K_m$ values of Ash1C and Ash1C-Mrg15 were determined for nucleosomes with 0.23 μM of S-[methl-$^3$H] adenosylmethionine. Data are plotted as mean + s.e.m. ($n = 2$). Student's t-test was used to determine significance (*$P < 0.05$). NS, not significant

displayed a positive correlation ($R = 0.58$, Fig. 3d). Moreover, the distribution profiles of Ash1 and Mrg15 were highly similar at common target genes (Fig. 3e, Supplementary Fig. 3e). Importantly, when we knocked down Ash1 and Mrg15 individually, their average intensity at Ash1 peaks displayed an interdependency (Fig. 3b, c, e, Supplementary Fig. 3e), further supporting that these two components interact with each other. We also noticed that Mrg15 occupancy is dependent on Ash1 at all Ash1 peaks regardless of Ash1 intensity, but Ash1 occupancy tended to be more dependent on Mrg15 at weak Ash1 peaks than strong Ash1 peaks (Fig. 3c, Supplementary Fig. 3f). These results suggest that Ash1 complex is recruited to its targets by a factor recruiting Ash1, and Mrg15's presence within Ash1 complex may help to stabilize the chromatin association of Ash1 complex.

Our ChIP-seq results revealed extensive Ash1 occupancy at the gene bodies of its target genes with an enrichment near the

transcription starting sites (TSSs) (Fig. 3e, Supplementary Fig. 3e, g), which is consistent with a previous report[47].

**Mrg15 facilitates Ash1 mediated H3K36me2 deposition**. Ash1 is a nucleosomal histone methyltransferase that catalyzes H3K36me2[12,18]. Because Mrg15 is a subunit of Ash1 (Fig. 1) that activates Ash1 (Fig. 2), we performed ChIP-seq experiments using antibodies against H3K36me2 in wild type and Ash1 knockdown and Mrg15 knockdown S2 cells. Because Mes-4 and Set2, the other two H3K36 methyltransferases, are responsible for global H3K36 methylation in *Drosophila*[48], we focused our analysis on the Ash1 target loci. As expected, the H3K36me2 level on Ash1 complex targeted loci (e.g., *vsx1* and *vsx2* genes) was almost completely abolished upon the depletion of Ash1 (Fig. 3e, Supplementary Fig. 3e). Meanwhile, the H3K36me2 level in non-Ash1 target regions (e.g., the *CG34435* gene) was not affected

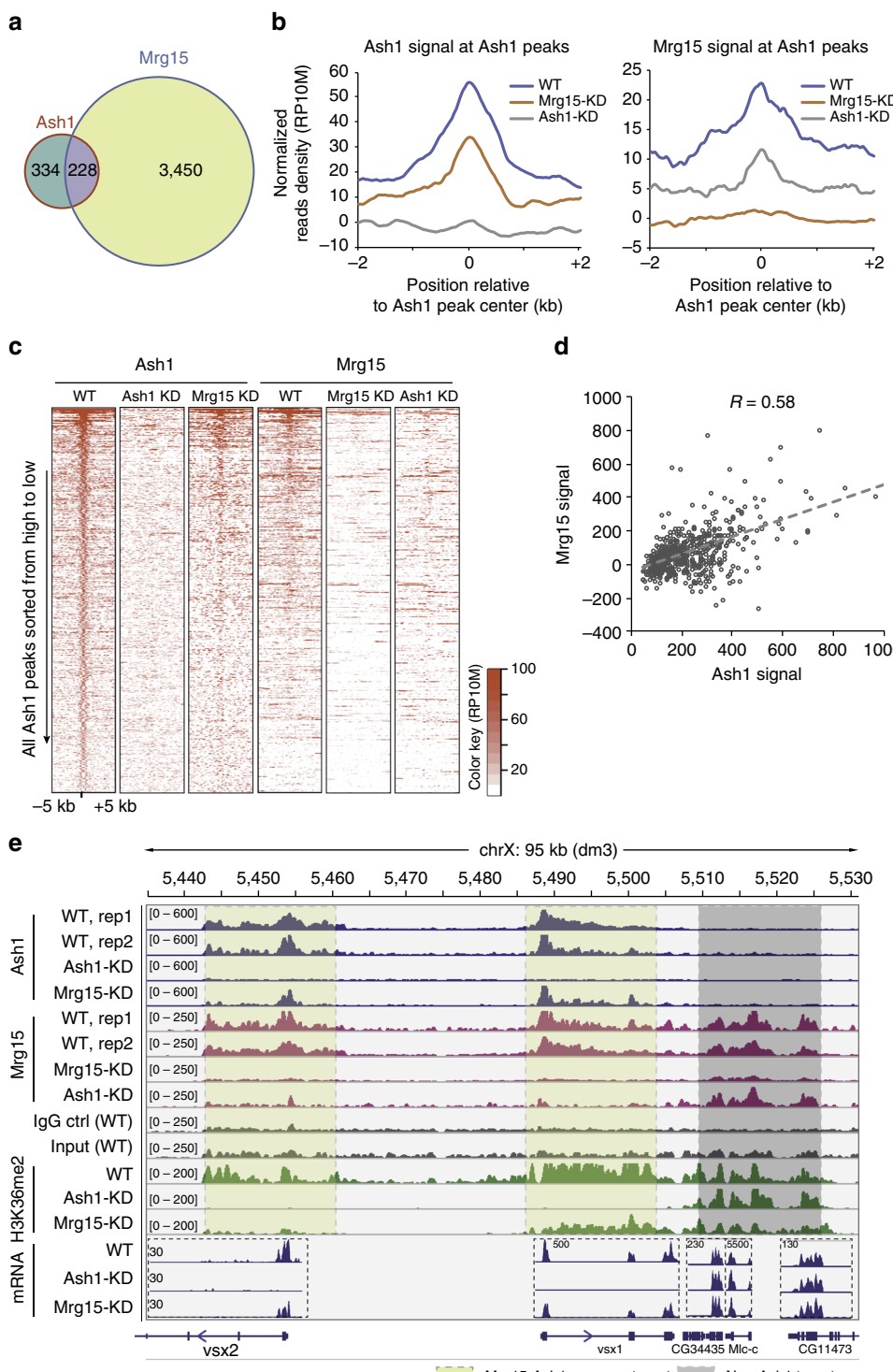

**Fig. 3** Ash1 co-localizes with Mrg15 and recruits Mrg15 to their common target genes. **a** Venn diagram show the overlap between Ash1 and Mrg15 peaks. **b** Profiles of average normalized read densities (RP10M) show the Ash1 (left panel) and Mrg15 (right panel) occupancy around Ash1 peaks in wild type and in Mrg15 knockdown cells. RP10M, reads per 10 million reads. **c** Heat maps show the occupancy of Ash1 and Mrg15 around Ash1 peaks in wild type, Ash1 knockdown and Mrg15 knockdown cells. Peaks are sorted according to Ash1 reads densities in wild-type cells. **d** Scatter plot shows the correlation between Ash1 and Mrg15 intensities at all Ash1 peaks. **e** Genome browser tracks show the binding of Ash1 and Mrg15, H3K36me2, and RNA-seq signals at several representative genes

(Fig. 3e). Importantly, depletion of Mrg15 resulted in a significant reduction of H3K36me2 at Ash1 target loci (Figs. 3e, 4a, b, Supplementary Fig. 3e, f), thus indicating the requirement of Mrg15 for the optimal deposition of Ash1-catalyzed H3K36me2.

These results are consistent with the in vitro activation of Ash1 mediated by Mrg15 (Fig. 2).

Because not all Ash1 peaks were equally enriched with Mrg15 (Fig. 3a, d), we divided the Ash1 peaks into two groups: Mrg15

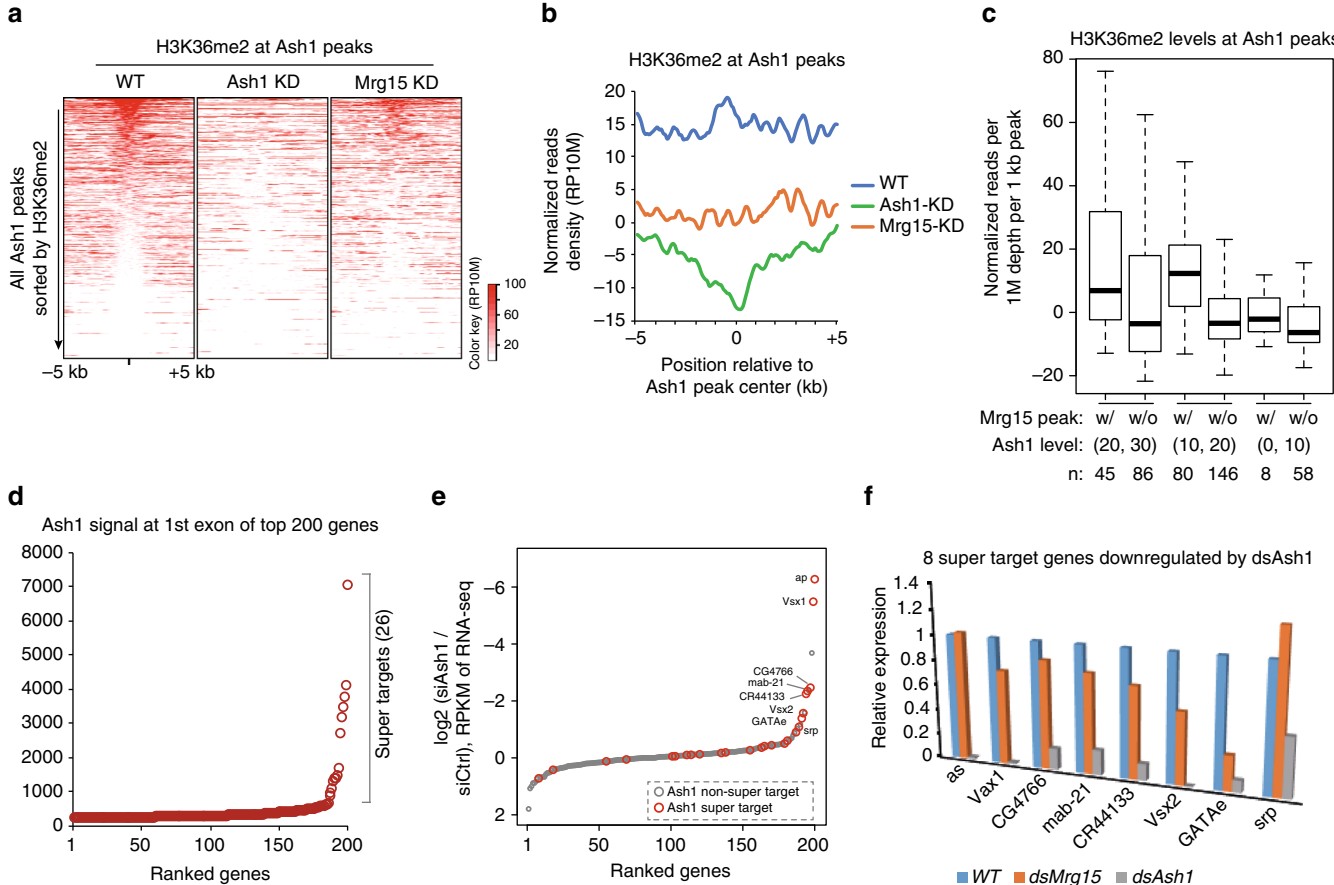

**Fig. 4** Mrg15 is required for proper deposition of Ash1 catalyzed H3K36me2 in S2 cells. **a** Heat maps show the H3K36me2 occupancy around Ash1 peaks in wild type, Ash1 knockdown and Mrg15 knockdown cells. All peaks were sorted according to H3K36me2 reads densities in wild-type cells. **b** Profiles of average normalized read densities (RP10M) show the H3K36me2 occupancy around Ash1 peaks in wild type, Ash1 knockdown and Mrg15 knockdown cells. **c** Boxplot of average normalized read densities (RPM per kb peak) of H3K36me2 occupancy at three subgroups of Ash1 peaks accompanied by high or low levels of Mrg15, respectively. **d** Scatter plot shows the distribution of Ash1 ChIP-seq signals at the first exons of the top 200 Ash1 target genes. Ash1 occupancy is not evenly distributed at these targets, with a subset of genes (the 26 super targets) containing exceptionally high levels of Ash1. **e** Scatter plot shows the mRNA level changes of the top 200 Ash1 target genes upon Ash1 knockdown. The 26 super target genes are marked in red. **f** The relative expression levels of 8 Ash1 top super target genes in wild type, Ash1 knockdown and Mrg15 knockdown cells

high and Mrg15 low. Then, we subdivided these two groups according to the Ash1 abundance and compared each pair that had a similar Ash1 abundance but different Mrg15 levels. In all cases, it was clear that the Mrg15 level contributed greatly to the level of H3K36me2 at Ash1 targets (Fig. 4c). These results further support an in vivo role of Mrg15 in the stimulation of Ash1-mediated H3K36me2 catalysis.

**Mrg15 contributes to the transcription of Ash1 target genes.** We then analyzed the effect of Ash1 and Mrg15 depletion on the expression of their target genes. In total, 398 genes were first defined as Ash1 target genes based on the presence of Ash1 peaks. Among them, 18 genes displayed a greater than twofold down-regulation and 8 genes displayed a greater than twofold upregulation after Ash1 knockdown in S2 cells based on RNA-seq results (Supplementary Data 1). Given that only a small fraction (18 out of 398) of these target genes became repressed after Ash1 depletion, we suspected that the majority of these genes may not be authentic Ash1 targets. Therefore, we selected top 200 Ash1 target genes and plotted normalized Ash1 ChIP-seq reads density at their first exons. Strikingly, 26 genes out from these 200 genes contained approximately half of the Ash1 reads (45%) after normalization, thus we defined them as Ash1 super target genes

(Fig. 4d). Then we plotted the fold change of RNA-seq signal and repressed genes were clearly enriched in the super target genes (Fig. 4e). We also noticed that although H3K36me2 level was reduced at both Ash1 super target and non-super target genes, H3K27me3 level was selectively increased at Ash1 super target genes (Supplementary Fig. 4), consistent with the biased tran-scriptional repression at Ash1 super target genes (Fig. 4e). On the other hand, Mrg15 knockdown in S2 cells displayed a much more modest transcriptional effect. Among the 8 top Ash1 super target genes that were repressed after Ash1 knockdown, 6 genes displayed various degrees of transcriptional downregulation after Mrg15 knockdown, although the levels of change was modest (Fig. 4f).

**Mrg15 is required for the TrxG protein function of Ash1.** Mrg15 is essential for Ash1 to achieve full catalytic activity in biochemical assays (Fig. 2) and in cultured cells (Fig. 4). However, Mrg15 downregulation only modestly repressed the expression of Ash1 target genes in S2 cells (Figs. 3e, 4f, Supplementary Fig. 3e). It was not clear whether such modest effect was due to insufficient knockdown or Mrg15's roles in other chromatin-modifying complexes. Therefore, we sought to explore the physiological significance of Mrg15-mediated Ash1 activation at the organism level.

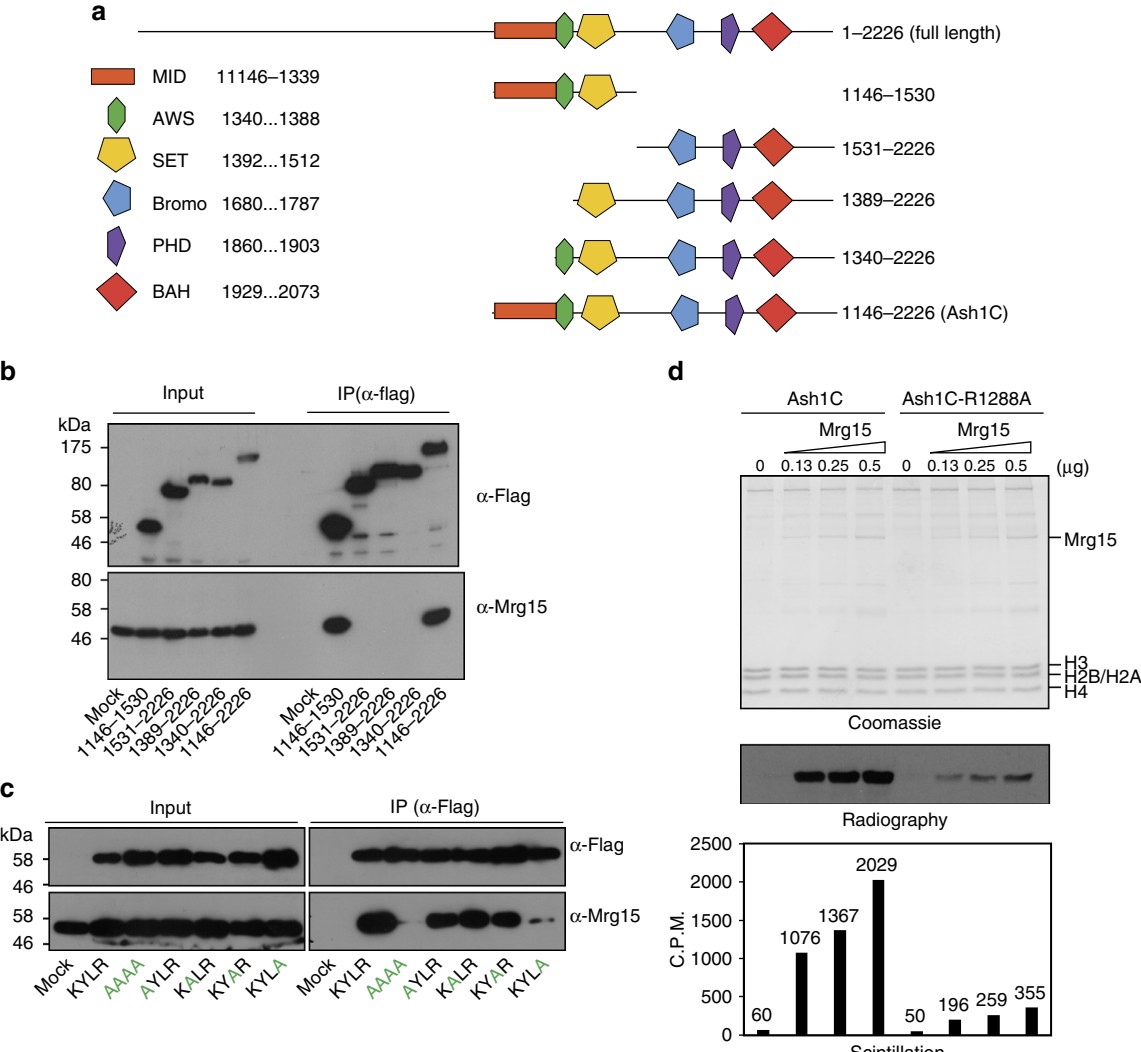

**Fig. 5** R1288 of Ash1 is critical for Mrg15 binding capacity and Mrg15-mediated Ash1 activation. **a** Schematic map shows the Flag-Ash1 C-terminal truncations used to determine the Mrg15 interaction domain. MID, Mrg15-interacting domain. **b** Co-IP results show Ash1 MID is essential for the Mrg15 interaction. **c** Co-IP results show that the R1288A mutation greatly weakens the interaction between Ash1C and Mrg15. **d** Histone methyltransferase assay results show that Mrg15-mediated Ash1 activation was severely compromised by the R1288A mutation

Mrg15 exists in multiple chromatin-modifying complexes and occupies more genomic regions than Ash1 (Fig. 3). Therefore, it is difficult to dissect the functional role of Mrg15 in the context of the Ash1 complex using loss-of-function studies because phenotypes caused by *mrg15* gene deletion could be attributed to the disruption of other Mrg15-containing complexes. Thus, we sought to assess the role of Mrg15 in the context of the Ash1 complex by generating an Ash1 line that displayed an attenuated Mrg15-Ash1 interaction.

We took advantage of the Co-IP experiments to map the protein region in Ash1 that was required for interaction with Mrg15. We tested a series of Flag-tagged Ash1C truncations and identified a short region ahead of the SET domain (amino acids 1146–1339), which we termed MID (Mrg15-interacting domain), that was required for Mrg15 binding (Fig. 5a, b). Ash1 MID is highly conserved among different species and contains an invariable KYLR motif (Supplementary Fig. 5). We performed alanine substitutions within this region and identified a single-amino acid substitution (R1288A) that greatly weakened the interaction between Ash1 and Mrg15 (Fig. 5c). Notably, this point mutation did not affect the interaction between Ash1 and Nurf55 because Nurf55 interacts with Ash1 at a different region (amino

acids 1314–2226 of *Drosophila* Ash1, corresponding to amino acids 2065–2964 of human Ash1L; Supplementary Fig. 6a).

Recombinant Ash1C (wild type) and Ash1C-R1288A (Mrg15 binding mutant) displayed similar basal enzymatic activity in vitro (Fig. 5d). However, unlike wild-type Ash1C, the Ash1-R1288A mutant protein failed to be robustly activated by the addition of Mrg15 (Fig. 5d). Thus, the introduction of R1288A mutation into Ash1 protein not only weakened the physical interaction between Ash1 and Mrg15 but also impaired Mrg15-mediated Ash1 activation in vitro.

To evaluate the physiological consequence of impairing Mrg15-mediated Ash1 activation, we generated an R1288A knock-in fly strain by Cas9/CRISPR technology. We first obtained an *ash1^R1288A/+* heterozygous strain and validated the DNA sequence of the two *ash1* alleles (Fig. 6a).

It has been reported that *ash1* loss-of-function strains exhibited several homeotic transformation phenotypes, including third leg to second leg transformation and haltere to wing transformation[17]. In wild-type flies, apical bristles exist on the second legs, but not on the third legs (Fig. 6b), and third leg to second leg partial transformation can be indicated by the presence of ectopic apical bristles on the third legs[17]. Similar to reported *ash1* loss-of-

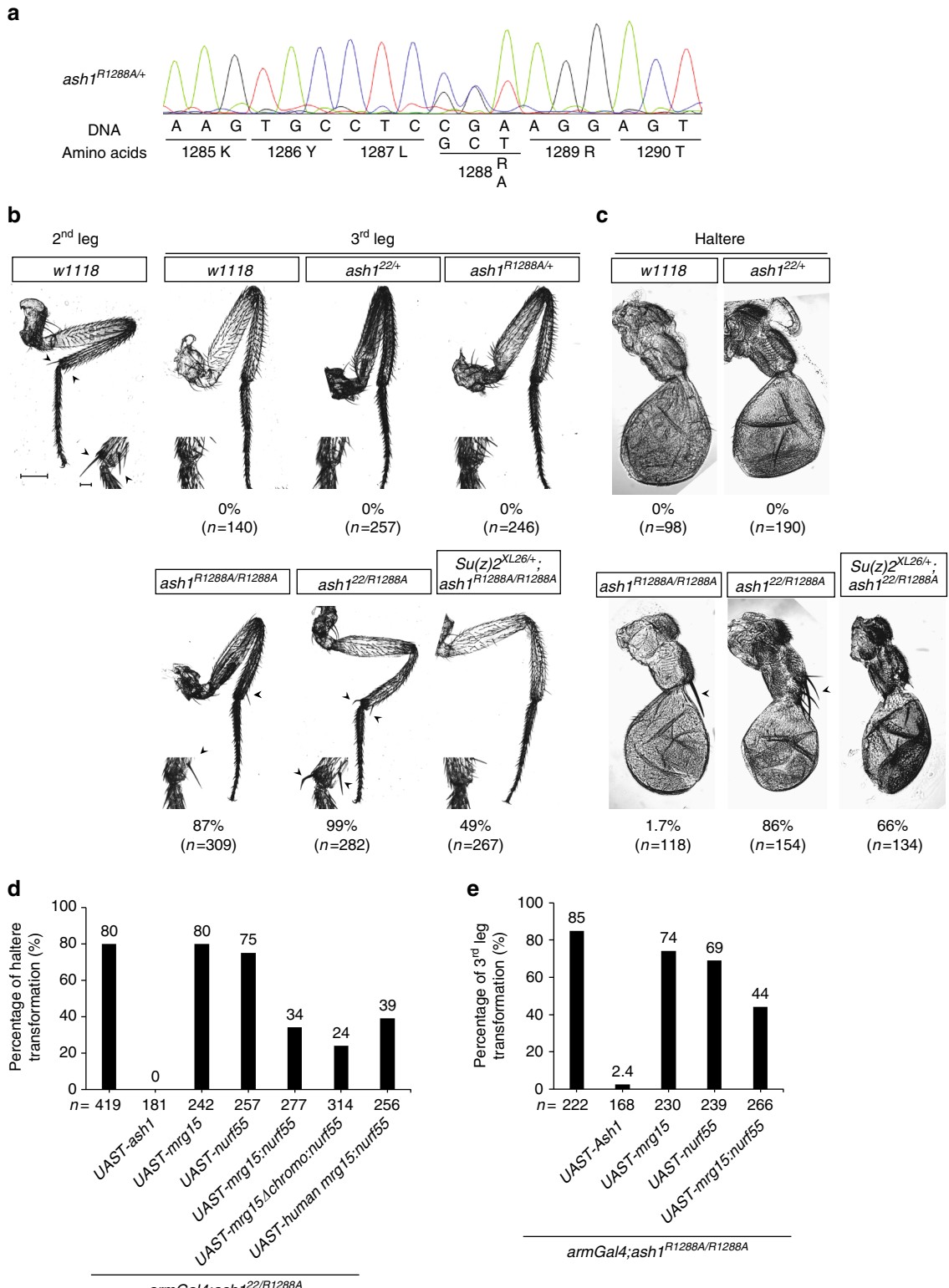

**Fig. 6** R1288A mutation compromised the TrxG protein function of Ash1. **a** Sanger sequencing result shows the genotype of *ash1*<sup>R1288A/+</sup> heterozygous fly. **b** Photographs and statistics show the third leg to second leg partial transformation phenotype in flies of indicated genotypes. The exact sample size (*n*) is indicated. Scale bars represent 50 μm (for the images of entire leg or haltere) and 10 μm (for the images of enlarged partial region of leg) respectively. Note that this transformation phenotype was observed in both *ash1*<sup>R1288A/R1288A</sup> and *ash1*<sup>22/R1288A</sup> flies, as indicated by the presence of ectopic bristles on the third leg, marked with arrowhead. **c** Photographs and statistics show the haltere to wing partial transformation phenotype in flies with the indicated genotypes. Note that the transformation phenotype was observed in both *ash1*<sup>R1288A/R1288A</sup> and *ash1*<sup>22/R1288A</sup> flies, as indicated by the presence of ectopic bristles on the haltere, marked with arrowhead. **d** Statistics show the percentage of haltere to wing partial transformation phenotype in flies with the indicated genotypes. **e** Statistics show the percentage of third leg to second leg partial transformation phenotype in flies with the indicated genotypes

function strains, 87% of $ash1^{R1288A/R1288A}$ mutant flies displayed third leg to second leg partial transformation, as indicated by the presence of the second leg specific apical bristle on the third leg (Fig. 6b). Moreover, the percentage of flies bearing ectopic apical bristles on their third legs was further increased to 99% in an $ash1^{22/R1288A}$ mutant strain ($ash1^{22}$ is an $ash1$ null allele caused by an early stop codon[17]). These results indicated that a reduction of Ash1 activity caused by impaired interaction with Mrg15 affected the function of Ash1 in the regulation of leg specification and that such effects could be increased by further reducing the Ash1 dose. Furthermore, introduction of one copy of $Su(z)2^{XL26}$ mutant, which is a null allele of both $Su(z)2$ and $Psc$ can partially suppress the third leg transformation phenotype of $ash1^{R1288A/R1288A}$ (Fig. 6b), suggesting that the homeotic transformation phenotype of $ash1^{R1288A/R1288A}$ was related to its compromised function in antagonizing Polycomb function.

In addition, we noted a low frequency (1.7%) of the haltere to wing partial transformation phenotype marked by the presence of ectopic bristles on the pedicel of halteres in $ash1^{R1288A/R1288A}$ mutant flies (Fig. 6c). Moreover, the frequency of the haltere to wing partial transformation phenotype was markedly increased (86%) in $ash1^{22/R1288A}$ mutant flies, which indicated that the optimal activity and dose of Ash1 is important for haltere specification. Like in third leg, introducing one copy of $Su(z)2^{XL26}$ mutant also partially relieve the transformation phenotype in haltere (Fig. 6c).

The above results established a connection between the attenuated Mrg15-Ash1 interaction and impaired TrxG protein function of Ash1. To further explore the functional significance of Mrg15 in this context, we generated transgenic flies overexpressing an Mrg15-Nurf55 fusion protein or other control proteins using the Gal4-UAST system. Because the interaction between Nurf55 and Ash1 is mediated by a different region of Ash1 (Supplementary Fig. 6a), the Mrg15-Nurf55 fusion protein can be tethered to Ash1-R1288A mutant protein.

As expected, overexpression of wild-type Ash1 fully rescued the haltere to wing transformation phenotype in $ash1^{22/R1288A}$ mutant flies, whereas overexpression of Mrg15 and Nurf55 had little effect (Fig. 6d), consistent with the biochemical observations that Nurf55 could not stimulate Ash1 activity (Fig. 2a) and Mrg15 displayed greatly attenuated stimulation towards Ash1C-R1288A (Fig. 5d). Importantly, overexpression of the Mrg15-Nurf55 fusion protein effectively rescued the above phenotype in more than half of the flies (Fig. 6d). Moreover, such rescue activity was independent of the chromo domain of Mrg15 because overexpression of the Mrg15Δchromo mutant fused with Nurf55 could also robustly rescue the same phenotype (Fig. 6d). This result is consistent with our earlier observation that the Mrg15 mutant without chromo domain can stimulate the catalytic activity of Ash1 in vitro (Fig. 2c). Finally, overexpression of an hMrg15-Nurf55 fusion protein also rescued the haltere to wing homeotic transformation phenotype (Fig. 6d), thus underscoring the functional conservation of Mrg15 in the context of Ash1 complex.

Next, we examined the rescue ability of Mrg15-Nurf55 fusion proteins in the more sensitive phenotype, third to second leg transformation. In the $ash1^{22/R1288A}$ background, this phenotype could only be rescued by overexpressing wild-type Ash1 (Supplementary Fig. 6b). However, in the milder $ash1^{R1288A/R1288A}$ background, which suffered less Ash1 activity loss than $ash1^{22/R1288A}$ background, overexpression of Mrg15-Nurf55 successfully restored the third leg ectopic apical bristles phenotype in approximately half of the flies (Fig. 6e).

Taken together, the above results demonstrated that the impaired TrxG protein function of the Ash1-R1288A mutant could be partially restored by stabilizing Mrg15 association,

further supporting a physiological role of Mrg15-mediated Ash1 activation.

It has been well documented that $ash1$ loss-of-function causes $Drosophila$ haltere and third leg homeotic transformation due to reduced UBX expression in $Drosophila$ haltere and third leg imaginal discs[22,49]. To test the role of Mrg15 in regulating UBX expression in the imaginal discs, we crossed $ash1^{22}/Tm3,actinGFP$ female and $ash1^{R1288A}/Tm3,actinGFP$ male flies, and then performed immunostaining with an antibody against UBX in the imaginal discs of the F1 third instar larvae. The genotypes of the GFP positive imaginal discs were either $ash1^{22/+}$ or $ash1^{R1288A/+}$ (Supplementary Fig. 7a), which displayed no homeotic transformation (Fig. 6). The genotype of the GFP negative imaginal discs was $ash1^{22/R1288A}$ (Supplementary Fig. 7a), which displayed the homeotic transformation phenotypes (Fig. 6). In $ash1^{22/+}$ or $ash1^{R1288A/+}$ larvae, UBX was highly expressed in the outer parts of the entire third leg imaginal discs (Supplementary Fig. 7b, c), but UBX expression was reduced in approximately half of the outer parts of the third leg imaginal discs of $ash1^{22/R1288A}$ larvae (long arrow heads in Supplementary Fig. 7b, c). These results support a role of Mrg15 in promoting UBX expression in third leg imaginal discs, which is consistent with the most obvious phenotype observed in mutant flies expressing Ash1-R1288A (Fig. 6).

## Discussion

The anti-repressive function of Ash1 as a TrxG protein was discovered many years ago[16,22]. However, the histone methyltransferase activity and substrate specificity were only clarified much later[12,18,21]. Despite the important role of Ash1 in the antagonization of PcG proteins and the maintenance of Hox gene expression, little is known about how its enzymatic activity is regulated, except for a report that the enzymatic activity of Ash1 is inhibited by H2A ubiquitination[13], which is a histone marker associated with Polycomb silencing. In this study, we identified Mrg15 and Nurf55 as integral subunits of the Ash1 complex, and characterized a role of Mrg15 in stimulating Ash1 enzymatic activity. However, many important questions remains to be answered, including how is Ash1 complex specifically recruited to its targets; how does Ash1 complex cooperate with other TrxG proteins, like Trithorax, Lid, and so on; what is the role of chromo domain of Mrg15?

The chromatin targeting of Ash1 complex remains an open question. Mrg15 contributes to Ash1's TrxG protein functional; however, it is clearly not the recruiter of Ash1. Although Mrg15 knockdown reduced Ash1 binding at its targets, such effect was relatively weak particularly at the Ash1 strong peaks (Fig. 3, Supplementary Fig. 3), suggesting that another factor is responsible for the initial recruitment of Ash1.

The cooperativity between Ash1 and other TrxG proteins, such as Trithorax and Lid are also interesting directions for future investigation. Although Trithorax was not detected in our purified Ash1 complex under the stringent condition that we applied (extensive washing with 500 mM KCl), Ash1 was reported to interact with Trithorax under less stringent conditions (binding and washing with 50 mM NaCl)[24] and Ash1's chromatin localization is dependent on Trithorax[47]. However, not all Trithorax targets are associated with Ash1[47], suggesting the existence of additional regulator(s) to be discovered. On the other hand, Lid, a TrxG protein that possesses histone H3K4 demethylase activity[50] interacts with Mrg15[45]. It would be interesting to examine whether such interaction may mediate a cooperativity between Lid and Ash1 complex.

Another interesting question is how broad does Ash1 complex regulate gene transcription. In the literature, this has been a

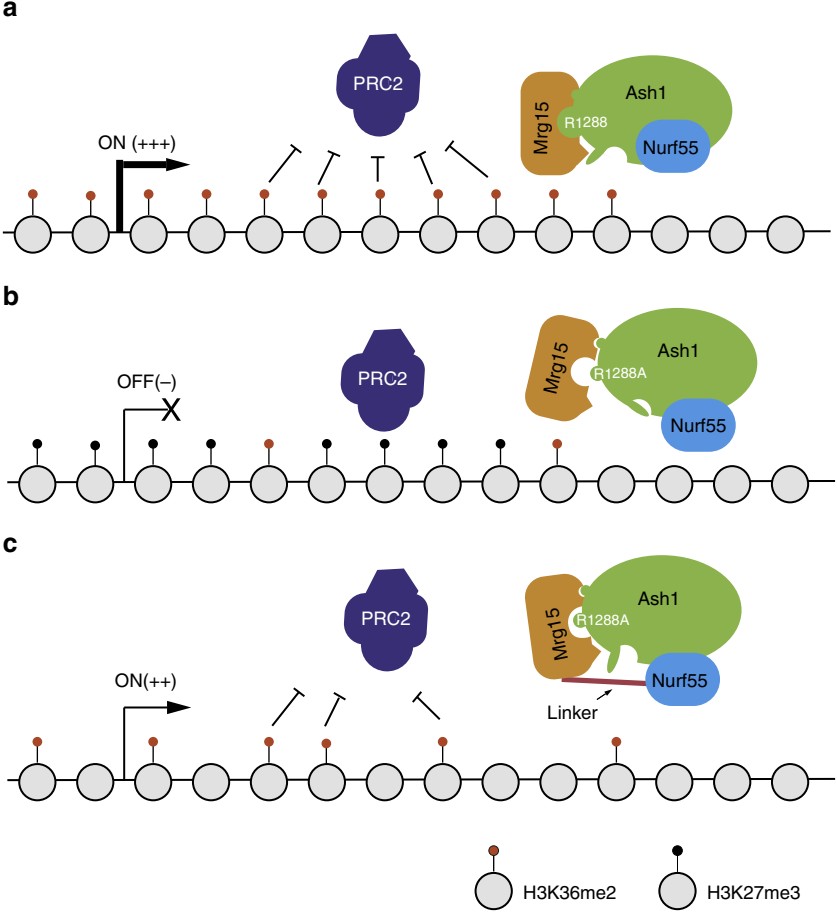

**Fig. 7** Model of the role of Mrg15 in Ash1 complex. **a** Wild-type Ash1 stably interacts with Mrg15 and becomes activated to produce adequate amounts of H3K36me2 that antagonizes Polycomb silencing at the common targets of Ash1 and Mrg15. **b** Ash1R1288A mutation greatly weakens its interaction with Mrg15 and impairs Mrg15-mediated activation, which causes the failure of anti silencing. **c** Expression of Mrg15-Nurf55 fusion protein partially restores the activation and anti silencing function of Ash1R1288A mutant

debate[27,47], our overall conclusion is that Ash1 affects the expression of only limited number of genes, and these genes are enriched in the set of Ash1 super target genes based on ChIP-seq intensity (Fig. 4d, e). This is in support to a previous report[47]. In S2 cells, the effect of Mrg15 knockdown at Ash1 target genes was relatively modest, and future investigations would answer whether this was attributed to incomplete knockdown and/or the mixed roles of Mrg15 in other chromatin modifying complexes. Nevertheless, Mrg15-mediated Ash1 activation is a physiologically relevant event supported by genetic evidences at the organism level (Fig. 6).

In previous studies, evaluations of the functional roles of Mrg15 and the yeast homolog Eaf3 were concentrated on the H3K36 methylation binding capacity. In budding yeast, genetic studies revealed that the Eaf3-containing HDAC Rpd3S complex functions downstream of H3K36 methyltransferase Set2 to suppress cryptic transcription initiation. Mutations affecting Set2 catalytic activity or Eaf3 H3K36 methylation binding activity led to histone hyperacetylation and internal transcriptional initiation at the gene bodies of active genes[42–44]. Our finding that Mrg15 without the chromo domain could activate the enzymatic activity of Ash1 (Fig. 2c, Supplementary Fig. 2c) represents biochemical evidence that Mrg15 is capable of modulating the catalytic activity of the associated histone-modifying enzyme. It will be very interesting to evaluate whether Mrg15 or Eaf3 may also regulate the catalytic activity of NuA4 histone acetyltransferase and/or the Sin3a/Rpd3S histone deacetylase. Notably, Eaf3 chromo domain

mutants resulted in increases of histone acetylation levels at the 3′ end of Rpd3S HDAC deacetylase complex target genes[42–44], and Eaf3 depletion compromised NuA4-catalyzed histone acetylation but not the chromatin recruitment of the NuA4 complex[51]. These results support a potential role of Mrg15 and Eaf3 in the biochemical regulation of the catalytic activities of other associated chromatin-modifying enzymes and provide highly interesting directions for future investigations.

The molecular mechanism by which Mrg15 activates Ash1 activity is another interesting question. Structural studies of SET domains of several closely related H3K36-specific methyltransferases, including NSD1 and hAsh1L, suggest that autoinhibition is an intrinsic folding characteristic of this subgroup of SET domain-containing histone methyltransferases[21,52–54]. A flexible loop from the post-SET subdomain blocks the access of the histone substrate from the catalytic groove. Although no evidence is yet available, it is attractive to speculate that incorporation of specific regulators can relieve the catalytic site from the self-blocked state to an accessible and catalytically competent configuration. In fact, a similar mechanism does exist in a different histone-modifying enzyme, PRC2. Autoinhibition was also observed in the SET domain of EZH2[55,56], whereas in the active PRC2 complex, extensive interactions were formed among complex subunits, which transforms the catalytic center of SET domain into an accessible and activated conformation[57–59]. Future structural analysis resolving the complex structure of Ash1-Mrg15 will help address this interesting question.

The purification of two independent human Ash1L complexes containing either hMrg15 or hMrgX proteins also offers interesting directions for future studies. Ash1L plays critical roles in mammalian development[19,60,61], and dysregulation of the human *ash1l* gene is linked to various diseases[62–66]. Inhibition of hAsh1L activity has been proposed as a potential therapeutic approach, particularly for tumors that are driven by *Hox* genes[67,68]. Further characterization of the regulatory roles of hMrg15 and hMrgX in activating hAsh1L may provide insights for rational designs targeting hAsh1L complexes. In particular, hMrg15 and hMrgX may function at different target genes, which might provide targeting specificity.

Finally, we propose a model to summarize the function of Mrg15 within Ash1 complex. In wide-type flies, Mrg15 stably interacts with Ash1 and stimulates its activity in generating H3K36me2 that antagonizes Polycomb silencing and maintains the transcriptional "ON" state of target genes (Fig. 7a). In flies bearing Ash1R1288A mutation, interaction between Mrg15 and Ash1 is attenuated, which impairs activation of Ash1 and compromises the anti silencing function of Ash1 (Fig. 7b). Expression of Mrg15-Nurf55 fusion protein in Ash1R1288A mutant flies artificially stabilizes Mrg15-Ash1R1288A association and partially restores the activation mechanism and the anti silencing function of Ash1 (Fig. 7c).

## Methods

**Antibodies**. The antibody against Flag (F1804, 1:1000 for western blot (WB)) and ANTI-FLAG M2 affinity gel (A2220) were purchased from Sigma. The antibody against RbAp46/48 was purchased from Cell Signaling Technology (4633S, 1:1000 for WB). The antibody against H3K36me1 was purchased from PTM BIO (PTM-623, 1:1000 for WB). The antibodies against H3K36me2 and H3K36me3 were purchased from Abcam (ab9049 and ab9050, 1:1000 for WB). The antibody against UBX was obtained from Developmental Studies Hybridoma Bank (FP6.87,1:300 for immunostaining). Polyclonal antibodies against *Drosophila* Ash1 (1:1000 for WB) and Mrg15 (1:5000 for WB) were produced in rabbits immunized with a recombinant *Drosophila* Ash1 truncated protein (441–880 aa) or a recombinant full-length *Drosophila* Mrg15 protein, and they were then immuno-purified with the corresponding antigens. Polyclonal antibodies against hMrg15 and hMrgX were produced by ABclonal Technology (1:1000 for WB). HRP-conjugated donkey anti-mouse and donkey anti-rabbit secondary antibodies were purchased from Pro-teintech (SA00001–8, SA00001-9, 1:10,000 for WB). Clean-Blot IP Detection Reagent (HRP) was purchased from Thermo Fisher (21230, 1:400 for WB).

**Co-Immunoprecipitation**. Endogenous Co-IP experiments were performed using the above described Ash1 and Mrg15 antibodies and also rabbit Normal IgG (Cell Signaling Technology, 2729S) as a control. Briefly, 2 μg antibodies were incubated with 2 mg S2 cell nuclear extracts in Buffer C (20 mM Tris-HCl, pH 8.0, 420 mM NaCl, 1.5 mM MgCl₂, 1 mM EDTA, and 10% glycerol) at 4 °C overnight. Then 100 μL protein G agarose (Millipore, 16–266) were added to capture the antibody-protein complex. After extensively washed with washing buffer (20 mM Tris-HCl, pH 8.0, 500 mM KCl, 1 mM EDTA, 10% glycerol, and 0.1% NP40), immunoprecipitated proteins were eluted using 1× SDS loading buffer and further anayzed by western blot.

**Cell culture**. *Drosophila* S2 cells were cultured with Schneidier's *Drosophila* medium (Gibco) plus 10% FBS (Gibco) at 28 °C. Human 293F cells were cultured with DMEM (Hyclone) plus 10% FBS (Gibco).

**Affinity purification and protein identification**. Nuclear extract (NE) prepared in Buffer C from about 100 mL of confluent cultured S2 cells or 10 × 100 mm-dishes of human 293F cells were used for batch purification. Anti-Flag M2 affinity gel (150 μL slurry) were equilibrited with Buffer C and then incubated with NE overnight at 4 °C. The resin was extensively washed with excess amounts of high stringency washing buffer (20 mM Tris-HCl, pH 8.0, 500 mM KCl, 1 mM EDTA, 10% glycerol, and 0.1% NP40). Proteins were eluted with 0.1 mg/mL Flag peptide. The elution were anayzled by SDS-PAGE and silver staining. Specific bands on gel were cut and proteins were resolved and digested with trypsin. The resulted peptides were analyzed by MALDI-TOF/TOF(MS/MS).

**Western blot**. For western blot analysis, protein samples were resolved by SDS-PAGE gels and transferred onto PVDF membranes. Then the membranes were incubated with appropriate antibodies at room temparature for 2 h or at 4 °C overnight followed by incubation with a HRP-conjugated secondary antibody. The

immunoreactive bands were visualized using Immobilon western chemilumines-cent HRP substrate (Millipore). Uncropped western blot scans with molecular weight reference were shown in Supplementary Figs. 8–11.

**Histone methyltransferase assay**. For the histone methyltransferase activity assay, a 30 μl reaction mixture containing 50 mM Tris-HCl (pH 8.0), 3 mM DTT, 1 mM MgCl₂, 230 nM S-[methyl-³H] adenosylmethionine (PerkinElmer), 0.02 nmol Ash1C and the amounts of titrated proteins indicated in the Figures, and 1 μg (or otherwise indicated) recombinant oligonucleosomes, were incubated at 30 °C for 4 h. The reaction was stopped by adding 4× SDS loading buffer and analyzed with SDS-PAGE, autoradiography and liquid scintillation.

For kinetic studies, reactions were performed at 30 °C for 30 min using titrated concentrations of recombinant nucleosomes or S-[methyl-³H] adenosylmethionine, as indicated. Standard errors were calculated from two replicates. The d.p.m data were calculated from the scintillation results, and kinetic regression curves were plotted using GraphPad Prism 5.

For histone methyltransferase activity assay followed by western blot detection, similar reaction system was adopted except that 1 mM S-adenosylmethionine and 2 μg recombinant oligonucleosomes were used in each reaction.

**ChIP-seq**. S2 cells were crosslinked with 1% formaldehyde for 10 min at room temperature and quenched by adding glycine to a final concentration G 0.125 M. The cells were washed with ice-cold PBS twice and then resuspended with buffer A (10 mM HEPES, pH 7.6, 10 mM EDTA, 0.5 mM EGTA, 0.25% Trithon X-100) and incubated for 10 min on ice. After centrifugation, the pellet was resuspended with buffer B (10 mM HEPES, pH 7.6, 1 mM EDTA, 0.5 mM EGTA, 200 mM NaCl, 0.01% Trithon X-100) and incubated for 10 min on ice. After centrifugation, the chromatin was resuspended in sonication buffer (20 mM Tris-HCl, pH 8.0, 140 mM NaCl, 1 mM EDTA) supplemented with 1% SDS and sonicated into fragments of 200–500 bp (Covaris M220) and insoluable material was removed by centrifugation of 14,000 r.p.m. for 10 min. The soluable chromatin was diluted by 10 times with binding buffer (20 mM Tris-HCl, pH 8.0, 500 mM NaCl, 1 mM EDTA) plus 0.1% Triton X-100 and incubated with antibody overnight at 4 °C. The chromatin-antibody complex were captured with Dynabeads Protein G (Life technology for 2 h. The beads were sequentially washed twice with binding buffer plus 0.1% Triton X-100, twice with binding buffer plus 1% Triton X-100, once with LiCl buffer (20 mM Tris-HCl, pH 8.0, 250 mM LiCl, 1 mM EDTA, 0.5% NP40, 0.5% sodium deoxycholate) and twice with TE (10 mM Tris-HCl, pH 8.0, 1 mM EDTA). The immunoprecipitated chromatin were eluted by incubating beads with elution buffer (0.1 M NaHCO₃, 1% SDS) at 55 °C for 2 h with vigorous vortex. DNA were purified after reversal of crosslink and deep sequencing libraries were constructed using NEBNext Ultra DNA Library Prep Kit and NEBNext Multiplex Oligos for Illumina.

**Fly stocks**. Flies were cultured on standard food medium at 25 °C. The strains used in this study were as follows: nos-Cas9 (attP2); ash1²² (BDSC #24161); arm-GAL4 (BDSC #1560); nos-phiC31; attP40 (BDSC #25709); and ash1^R1288A, which was generated by CRISPR/Cas9 using 5′-cggaggtacttttcgcaata-3′ gRNA and a 2 kb genomic DNA fragment with the R1288 site at the middle inserted into pEasyT vector as donor plasmid and R1288A mutation was generated by Quickchange. Ash1^R1288A gRNA and donor plasmids were purified using a Qiagen Plasmid Midi Kit (#12145) and introduced into nos-Cas9 embryos using standard procedures. To make the UAST-ash1, UAST-mrg15, UAST-nurf55 and UAST-mrg15:nurf55 constructs, fragments were subcloned into an attB-UAST vector using ClonExpress II One Step Cloning Kit (Vazyme, C112). All plasmids were verified by DNA sequencing. The plasmid DNA was introduced to nos-phiC31;attP40 embryos using a standard procedure to generate transgenic flies.

**Homeotic transformation phenotype analysis and microscopy**. For each genotype, 100–300 flies of 3 day to 5 day old were used for homeotic transformation analysis. For representing images acquisition, the haltere and leg were manually dissected from 3-day to 5-day-old flies and images were obtained using a Zeiss Imager M1 microscope.

**Imaginal disc immunostaining**. The ash1²²/Tm3,actinGFP females and ash1^R1288A/Tm3,actinGFP male flies were crossed and the F1 third instar larvae were collected and dissected in ice-cold PBT (10 mM NaH₂PO₄/Na₂HPO₄, 175 mM NaCl, pH 7.4, and 0.1% Triton X-100). The carcasses with imaginal discs attached were fixed with 4% formaldehyde in PBT for 15 min at room temperature. After washing three times with PBT and blocking with 5% normal goat serum in PBT for 1 h, the samples were incubated with mouse anti-UBX antibody (1:300, DSHB, FP6.87) overnight at 4 °C. The secondary antibody, Alexa-568-conjugated goat anti-mouse (1:300, Molecular Probes), was incubated for 1 h at room temperature. The samples were also stained with 0.1 mg/ml DAPI (4′,6′-diamidino-2-phenylindole; Sigma) for 5 min. The wing, haltere and third leg imaginal discs were isolated from carcasses and mounted on microscope slides for image analysis. Images were obtained using a ZEISS LSM 700 microscope.

**Bioinformatic analysis of high-throughput sequencing data.** After ChIP libraries were constructed, single-end sequencing was performed on the Illumina GA II or HiSeq 4000 platforms. Sequencing reads were aligned to the UCSC *dm3 Drosophila* genome sequence, and PCR duplicates were removed. Read densities were counted using IGV tools with read lengths extended to the average fragment size, followed by normalization to a sequencing depth of one million for each sample. Ash1/Mrg15 binding peaks were called by MACS software (v2.0.10) with parameters "--nomodel --extsize 250 -q 0.1 --broad-cutoff 0.1 –broad". To define Ash1 super target genes, we sorted all genes by Ash1 ChIP-seq reads density within the first exon (exons shorter than 200 bp were extended to 200 bp), and all values were normalized to the sequencing depth of 10 M and region length of 1 kb. H3K36me2 read densities at Ash1/Mrg15 peaks were counted based on uniquely mapped ChIP-seq reads and normalized to a 1 kb peak length, which was similar to RPKM in RNA-seq analysis. For RNA-seq, total RNA was isolated by the TRIzol Reagent (Life Technologies). mRNA with a poly(a) tail was purified and subjected to sequencing library constructions. Reads were mapped to the UCSC *dm3 Drosophila* genome with TopHat (v1.4.1). The expression levels were quantified with Cufflinks (v2.0.2) software.

**Data availability.** All high-throughput sequencing data have been deposited under the GEO accession number GSE93100.

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

## Acknowledgments

We thank Dr. Gerd A. Blobel for providing the human Ash1L cDNA. We thank Dr. Hongjie Zhang for technical support in handling radioactive isotopes. We thank Dr. J. Müller for discussion. This work was primarily supported by the China Natural Science Foundation (grant 31530037). This work was also supported by grants from the Chinese Ministry of Science and Technology (2015CB856200 and 2016YFA0100400), the China Natural Science Foundation (31571344, 31521002, and 31425013), the Strategic Priority Research Program (XDB08010103, to B.Z.) and the Youth Innovation Promotion Association (2017133, to Z.Z.) of the Chinese Academy of Sciences.

## Author contributions

C.H. conducted the purification, biochemistry and ChIP-seq experiments. F.Y. and C.H. performed the *Drosophila* phenotype analysis and disc immunostaining experiments. Z.Z. analyzed the ChIP-seq and RNA-seq data. G.C., L.L., and S.C. conducted the Mass Spec analysis. J.Z. and Y.Z. assisted in generation of polyclonal antibodies. B.Z. and R.X. supervised the experiments and analyzed the data. C.H., Z.Z., and B.Z. wrote the manuscript and all the authors read and commented on the manuscript.

## Additional information

**Competing interests:** The authors declare no competing financial interests.

