## [Peer Review File · Nature Communications]

Reviewers' comments:

Reviewer #1 (Remarks to the Author):

Review of NCOMMS-17-11064 by Dr Zhu and co-workers entitled "Mrg15 stimulates H3K36 methyltransferase activity of Ash1 and facilitates Trithorax group protein function of Ash1 in *Drosophila*".

Ash1 is an histone methyl transferase that catalyse the H3K36me3 modification. It belongs to the Trithorax group protein. Despite accumulating studies in the last years, the molecular mechanisms regulating this histone modifier are still lacking. In the present study, Huang and co-workers combine affinity purification to identify regulators of Ash1 activity. By means of in-vitro (Histone methyltransferase assays) and in-vivo (siRNA, ChIP-seq and RNA-seq and point mutation in flies) assays, they characterize the role of two newly identified partners of Ash1; Mrg15 (MOF related gene) and Nurf55 (that forms complexes with several chromatin modifiers such as PRC2 or CAF-1 complex...).

The paper is technically sound and the combination of the various technique is well thought. We tough suggest submitting a revised version of the present study.

Related to Fig.1

Huang et al., generated S2 cells expressing an epitope tagged Ash1 to purify Ash1 associated complex followed by mass spectrometry analysis of associated partners (Fig. 1b and suppl Fig1a). The interactions are further validated by co-IP / western-blot (1c and sup 1b). The purification in fig 1b is performed from a truncated form of Ash1 (Ash1-C) although a purification has been, as well, performed from the full length Ash1 (Sup 1a). Most of biochemical assays (Fig2, Fig5) are being performed using the truncated form.

1. The authors also included two purifications from human hAsh1 (HEK cells – 1d, 1e and sup 1c). These results support the idea of a complex between these 3 proteins is conserved from *Drosophila* to human. Although of a great relevance, the preliminary work on human hAsh1-I does not improve the overall conclusions from the present paper (no comparison with dAsh1 in HMT, no ChIP-seq nor RNA-seq data, almost absent from in vivo part), and would really benefit from further experiments. We would therefore suggest to move those data in a supplementary figure.

2. Ash1 interacts with Trx (Rozovskaia et al, MCB, 1999). Is this interactions recovered in the context of Ash1-C and full-length Ash1? It has been previously reported that Mrg15 is part of the Tip60 complex in *Drosophila* (Kush et al, science 2004) as well as Sin3 complex. Nurf55 is part of several chromatin complexes? Are these interactions part of Ash1-C and full length Ash1 complex?

3. Are the expression levels of Flag-Ash1-C and full-length Flag-Ash1 comparable to endogenous Ash1 level?

4. Endogenous Ash1 co-IP as well as endogenous Mrg15 co-IP should be included in fig.1 (rather than sup fig1c) since these experiments would clearly demonstrate an interaction between endogenous proteins. Yet, it is not clear whether Ash1 co-immunoprecipitate with Mrg15 (loading issue or reflecting a low amount of Ash1 bound to Mrg15?).

Minor comment: numbering lanes and indicating the input percentage would help to reader to understand the results.

Related to Fig.2

The authors further tested the histone methyltransferase activity of Ash1-C in vitro. They observed that Mrg15 increases Ash1 activity on RON (recombinant oligonucleosomes) as a substrate (Fig2a) independently of Mrg15 chromodomain (Fig2c). This increase in Ash1 activity is not observed in the presence of Nurf55 (Fig2a).

5. Do Nurf55 and Mrg15 improve Ash1-C binding to RON? Is this dependent on Mrg15 chromodomain?

6. Does Mrg15 change Ash1-C specificity toward H3K36me1 /2 /3? Is Mrg15 chromodomain involved in Ash1-C specificity?

Related to Fig.3 and 4

Next, Huang and co-workers mapped Ash1 and Mrg15 chromatin binding sites genome wide (ChIP-seq) after knock-down of one or the other subunit of the complex (Fig3b); H3K26me2 profiles are also mapped. They observe common peaks for both proteins, supporting their interactions (as observed in Fig.1) as well as independent peaks (Fig3a). They further analyze Ash1 and Mrg15 occupancy at common peaks after KO (Fig3c.d.e.f). They concluded that Mrg15 is not required for Ash1 recruitment to chromatin (Fig.3b and c) but essential for H3K36me2 deposition (Fig.3b and 4) since KD doesn't affect the expression level of other subunits of the complex (Sup Fig.3b).

7. The authors focus their analysis on Ash1/Mrg15 common peaks (expect Fig.4c) to understand the function of Mrg15 associated with Ash1.

8. The author should also include a heat map containing the 60% of Ash1 peaks that doesn't overlap with Mrg15 and clarify whether those peaks are also affected by Mrg15 depletion.

9. Is H3K36me2 deposition affected in Mrg15 KD? An heat map would also help in this case

10. Given the central role of H3K36me2 in transcription elongation, and the recent discovery of Mrg15 role in mRNA splicing (Luco & al Science, 2010, Gonzales et al, NSMB, 2014), the authors should provide a global view of mRNA expression level in Ash1 – Mrg15 KD cells. Ideally log2FC RNAseq (WT Vs KD) heat maps should be imposed over Ash1 heat maps centered on peaks (by imposing K-means clustering). Or, if difficult to visualize, a box plot measuring RNAseq log2FC (WT Vs KD) at genes right-on or close-to a given Ash1 peak, in both KDs.

Related to figure 5

The authors identified the MID domain of Ash1 and more specifically the Arg R1288 as a critical residue for Mrg15 interaction (Fig 5b, c). They also observed a reduced activity of Ash1-c R1288 as compare to Ash1-c.

11. Minor comment: Fig 5a – It would help the reader to present Ash1 truncations carton diagram ordered as loaded in Fig5b.

Reviewer #2 (Remarks to the Author):

In this manuscript, Huang et al identified the novel Ash1 complex with Mrg15 and Nurf55 in *Drosophila* using S2 cells stably expressing Flag-tagged truncated Ash1C, and found that Mrg15 stimulates the catalytic activity of Ash1 in vitro. Further, in vivo, Mrg15 and Ash1 are recruited to the common target gene sites and Mrg15 is required for H3K36me2 around these sites. They also identified an interesting point mutant Ash1-R1288A, which abrogates its interaction with Mrg15, and its knock-in flies display multiple homeotic transformation phenotype, indicating the interaction of Ash1 with Mrg15 is critical for Ash1 function in fly.

These findings are novel, highly original and informative and interesting to people working in not only TrxG-PcG fields but also general chromatin remodeling research. The methods they employed, such as Flag IP/mass spectrometry, ChIP-seq and data analysis by bioinformatics are straightforward, well standardized and well presented. Considering the points above, this reviewer suggests this manuscript is acceptable for the Journal. However, minor comment, albeit trivial, is as below.

Minor comment

On lines 69 and 150, the authors described "Mrg15 mediates allosteric activation of Ash1" or "via an allosteric activation mechanism". They also presented kinetic data in Fig2. E, showing Vmax is

robustly activated albeit the K_m for nucleosomes are affected very small. This is in contrast to the concept of classical enzymology; hyperbolic saturation in Michaelis-Menten model vs sigmoidal allosteric model. Classical enzymology teaches us allosteric activation significantly increases the affinity for substrate, i.e. decreases the K_m , without much effect on the turnover V_{max} of enzyme. If authors stick to the term "allosteric", they had better reconsider this comment.

Reviewer #3 (Remarks to the Author):

The molecular nature of the Ash1 complex and its biochemical activity have been slow to be characterized, after several false starts that resulted in incorrect claims. It is clear now that it is a histone H3K36 methyltransferase that acts in concert with Trithorax as an antagonist to Polycomb repression and promotes transcription. More complex is the question of the molecular nature of Ash1 and how it is recruited. The present paper makes significant progress in characterizing the protein complex associated with Ash1. Affinity purification of FLAG-tagged truncated, full length Ash1 and human Ash1L, followed by mass spectrometry showed that the Ash1 complex includes Mrg15 and Nurf55 or the human orthologues of these proteins. The paper shows very competently that these components have no intrinsic methyltransferase activity but that Mrg15, with or without its chromodomain, greatly stimulates the catalytic activity of Ash1. This is an important advance that will be relevant to all who study Polycomb repression, its regulation, Trithorax, Ash1 and associated mechanisms in development and disease.

A few comments:

p.7 line 138. Figure 2A,B,C. It appears from the figure that Mrg15 delta chromodomain is substantially less potent at stimulating Ash1 catalytic activity. This impression from the autoradiography is in part corrected by the scintillation counts plotted below. It would be useful to clarify this point.

p. 8. Given its involvement in other complexes, it is not surprising that Mrg15 binds to many genomic sites in the absence of Ash1. At sites common to Ash1 and Mrg15, the authors show that Ash1 recruits Mrg15. However, more than half of the Ash1 binding peaks in the genome are unaccompanied by Mrg15. We don't know how Ash1 is recruited and although Ash1 can further recruit Mrg15, it is clearly not sufficient: some other signal or component is required. Although the authors have compared the genomic distributions of Ash1, Mrg15 and transcriptional activity, they tell us very little about them. Figure 3B and Suppl. Figure 3C show the RNA tracks for only part of the region indicated in the figures. Are all genes that have Ash1 + Mrg15 transcriptionally active? Are genes that have Ash1 alone still active?

p. 9 line 179. This is incorrect. The major H3K36 methyltransferase in Drosophila is Set2. There is a gene named NSD (Nuclear receptor binding SET domain protein), also known as Mes-4 or dMes-4 (after the *C. elegans* Mes-4) which also encodes a Set2-like methyltransferase but it is not the main H3K36 methylase.

p. 9 line 182. H3K36me2 is almost abolished upon depletion of Ash1. Since Ash1-binding genes are presumed to be transcriptionally active, this implies that Ash1 excludes the normal Set2 activity. How is this explained?

Suppl. Figure 3C. it is interesting to note that H3K36me2 virtually disappears over the Grip gene upon Ash1 knockdown although this gene normally binds little or no Ash1. Although it is possible that Grip expression depends on some unknown Ash1 target gene, it would be useful to know the transcription status of this gene, which is apparently not given in the figure.

There have been claims that Ash1 affects transcription more globally than just at the sites of the

major peaks. It would be very interesting to give some more attention to these results. Figure 3b and Suppl. Figure 3c show only the part of the data corresponding to common peaks of Ash1 and MRG15. More discussion of the relationship of Ash1 to transcription is needed.

p. 11 line 212. This is an excellent approach.

p. 13, line 264. A sentence of explanation would be useful

The discussion is a brave attempt but disappointing. The paper makes important progress in giving us a clear idea of the components of the Ash1 complex and their effects on catalytic activity but the key questions remain unanswered and mysterious. How is Ash1 recruited? How does it antagonize Polycomb repression? Is it really through H3K36me2 inhibition of PRC2? If so, why does the H3K36 methylation produced by Set2 not sufficient? How is Ash1 related to Trx? These are some of the key unanswered questions. The discussion has little to say about them. It could be abbreviated and let the data speak for themselves.

Point-by-point response to reviewers' comments:

Reviewer #1 (Remarks to the Author):

Review of NCOMMS-17-11064 by Dr Zhu and co-workers entitled "Mrg15 stimulates H3K36 methyltransferase activity of Ash1 and facilitates Trithorax group protein function of Ash1 in Drosophila".

Ash1 is an histone methyl transferase that catalyse the H3K36me3 modification. It belongs to the Trithorax group protein. Despite accumulating studies in the last years, the molecular mechanisms regulating this histone modifier are still lacking. In the present study, Huang and co-workers combine affinity purification to identify regulators of Ash1 activity. By means of in-vitro (Histone methyltransferase assays) and in-vivo (siRNA, ChIP-seq and RNA-seq and point mutation in flies) assays, they characterize the role of two newly identified partners of Ash1; Mrg15 (MOF related gene) and Nurf55 (that forms complexes with several chromatin modifiers such as PRC2 or CAF-1 complex...).

The paper is technically sound and the combination of the various technique is well thought. We tough suggest submitting a revised version of the present study.

Related to Fig.1

Huang et al., generated S2 cells expressing an epitope tagged Ash1 to purify Ash1 associated complex followed by mass spectrometry analysis of associated partners (Fig. 1b and suppl Fig1a). The interactions are further validated by co-IP / western-blot (1c and sup 1b). The purification in fig 1b is performed from a truncated form of Ash1 (Ash1-C) although a purification has been, as well, performed from the full length Ash1 (Sup 1a). Most of biochemical assays (Fig2, Fig5) are being performed using the truncated form.

1. The authors also included two purifications from human hAsh1 (HEK cells – 1d, 1e and sup 1c). These results support the idea of a complex between these 3 proteins is conserved from Drosophila to human. Although of a great relevance, the preliminary work on human hAsh1-I does not improve the overall conclusions from the present paper (no comparison with dAsh1 in HMT, no ChIP-seq nor RNA-seq data, almost absent from in vivo part), and would really benefit from further experiments. We would therefore suggest to move those data in a supplementary figure.

We have moved the human Ash1L purification data into supplementary figures as suggested (new Supplementary Fig. 1c).

2. Ash1 interacts with Trx (Rozovskaia et al, MCB, 1999). Is this interactions recovered in the context of Ash1-C and full-length Ash1?

We purified Flag-Ash1C and full-length Flag-Ash1 complexes under stringent conditions (binding with 420 mM NaCl and extensive washing with 500 mM KCl plus 0.1% NP40), and Trx was not identified in the elution by mass spectrometry analysis. This is consistent with the knowledge that Ash1 or Ash1L has not been recovered from the purification of drosophila Trx or mammalian MLL1 or mammalian MLL2 complexes (MLL1 and MLL2 are the mammalian orthologues of drosophila Trx). In the abovementioned MCB paper (cited

as ref. 44), the authors used a candidate approach and directly performed Co-IP experiments for Trx and Ash1 using a much lower stringency (binding and washing with **50 mM NaCl**). Thus, our conclusion is that Trx is not an integral subunit of Ash1 complex, although it may transiently interact with Ash1 complex. We added a paragraph into the discussion to discuss this.

It has been previously reported that Mrg15 is part of the Tip60 complex in *Drosophila* (Kush et al, science 2004) as well as Sin3 complex. Nurf55 is part of several chromatin complexes? Are these interactions part of Ash1-C and full length Ash1 complex?

Yes, Mrg15 interacts with Tip60 complex and is also a core subunit of Sin3 complex (cited as ref. 67-75). And Nurf55 and its mammalian orthologues (RbAp46/48) are core subunits of many chromatin complexes, including PRC2, NURF, CAF1, NuRD, and HJURP complexes (cited in ref 5-8, 48-55). Specific subunits of the abovementioned complexes were not identified in our affinity purified materials with Flag-Ash1C, full length Flag-Ash1, and Flag-hAsh1L-C. These results support that Mrg15, Nurf55 and Ash1 compose the Ash1 core complex, but this does not exclude the possibility that under certain biological context these different Mrg15-containing complex may work in proximity on chromatin.

3. Are the expression levels of Flag-Ash1-C and full-length Flag-Ash1 comparable to endogenous Ash1 level?

It is difficult for us to directly compare the expression level of Flag-Ash1-C with the endogenous Ash1, because our antibody against Ash1 recognizes the N-terminal region of Ash1 that is missing in Flag-Ash1-C. But we were able to test the expression levels of endogenous Ash1 and full-length Flag-Ash1 by western blot. It is quite clear that the full length Flag-Ash1 is overexpressed for at least several fold (new Supplementary Fig. 1a).

Nonetheless, we think this does not affect our analysis, because the overexpression cell line was only used in the affinity purification experiments. Endogenous Mrg15 and Nurf55 proteins are much more abundant than endogenous Ash1, because they are also subunits of other chromatin complexes. More over, the interaction between endogenous Ash1 and Mrg15 was verified by Co-IP experiments (Fig. 1d) and ChIP-seq experiments showed an Ash1-dependent localization of Mrg15 at their common targets (new Fig. 3b, 3c). In addition, for the assessment of Mrg15's regulatory role within Ash1 complex at the organism level, Ash1-R1288A mutation was introduced into the endogenous locus of *Drosophila* by knock-in experiments (Fig. 6).

Endogenous Mrg15 co-IP should be included in fig.1 (rather than sup fig1c) since these experiments would clearly demonstrate an interaction between endogenous proteins. Yet, it is not clear whether Ash1 co-immunoprecipitate with Mrg15 (loading issue or reflecting a low amount of Ash1 bound to Mrg15?).

Minor comment: numbering lanes and indicating the input percentage would help to reader to understand the results.

We agree with the reviewer and have moved these results into the main figures (new Fig. 1d) and labeled the input percentage as suggested.

Related to Fig.2

The authors further tested the histone methyltransferase activity of Ash1-C in vitro. They observed that Mrg15 increases Ash1 activity on RON (recombinant oligonucleosomes) as a substrate (Fig2a) independently of Mrg15 chromodomain (Fig2c). This increase in Ash1 activity is not observed in the presence of Nurf55 (Fig2a).

5. Do Nurf55 and Mrg15 improve Ash1-C binding to RON? Is this dependent on Mrg15 chromodomain?

The addition of Nurf55 didn't alter the catalytic behavior of Ash1C (Fig. 2a), suggesting that it has little effect on improving substrate binding, because both enhanced binding or increased turnover will stimulate the enzymatic activity.

For Mrg15, the K_m value of Ash1C-Mrg15 complex was modestly reduced compared with Ash1C alone, but the difference was not statistically significant ($P = 0.79$, Student *t*-test), suggesting that Mrg15's effect on Ash1C's binding to RON is limited. In contrast, V_{max} of Ash1C-Mrg15 was greatly increased compare to Ash1C alone, suggesting a much improved turnover rate of the enzyme. We added the above *P* values into new Fig. 2e.

6. Does Mrg15 change Ash1-C specificity toward H3K36me1/2/3? Is Mrg15 chromodomain involved in Ash1-C specificity?

The catalytic specificity of Ash1C and Ash1C-Mrg15 was analyzed by western blot using antibodies specific for H3K36me1/2/3, respectively. Mrg15 effectively stimulated the production of H3K36me1 and H3K36me2, but did not generate detectable H3K36me3. Thus our conclusion is that Mrg15 stimulates the overall enzymatic activity of Ash1, but does not change the product specificity. Given that the full length Mrg15 does not appear to have a role in altering the specificity of Ash1, its chromodomain is unlikely to play a role in altering Ash1 specificity.

Related to Fig.3 and 4

Next, Huang and co-workers mapped Ash1 and Mrg15 chromatin binding sites genome wide (ChIP-seq) after knock-down of one or the other subunit of the complex (Fig3b); H3K26me2 profiles are also mapped. They observe common peaks for both proteins, supporting their interactions (as observed in Fig.1) as well as independent peaks (Fig3a). They further analyze Ash1 and Mrg15 occupancy at common peaks after KO (Fig3c.d.e.f). They concluded that Mrg15 is not required for Ash1 recruitment to chromatin (Fig.3b and c) but essential for H3K36me2 deposition (Fig.3b and 4) since KD doesn't affect the expression level of other subunits of the complex (Sup Fig.3b).

7. The authors focus their analysis on Ash1/Mrg15 common peaks (except Fig.4c) to understand the function of Mrg15 associated with Ash1.

8. The author should also include a heat map containing the 60% of Ash1 peaks that doesn't overlap with Mrg15 and clarify whether those peaks are also affected by Mrg15 depletion.

We are grateful to this suggestion. During revision, we analyzed all the Ash1 peaks and realized that good amounts of Mrg15 were present at many Ash1 peaks that were not called as Mrg15 peaks in Fig. 2a. This is likely due to the arbitrary cut off during peak calling.

Therefore, we decided to perform more bioinformatics analysis. Our new observations are summarized as the following: 1. At all Mrg15 peaks, Ash1 is not apparently enriched, likely due to Mrg15's presence in multiple chromatin modifying complexes and Ash1 complex only contributes to a small fraction of Mrg15's chromatin occupancy (new Supplementary Fig. 3c); 2. At the majority of Ash1 peaks, Mrg15 is clearly enriched and such enrichment is Ash1-dependent (new Fig. 3b, 3c, Supplementary Fig. 3d); 3. In the previous version, we mentioned that Ash1 occupancy was slightly reduced upon Mrg15 knockdown, during revision, we realized that such effect was stronger at weak Ash1 peaks. Therefore we performed more analysis and included them as new Fig. 3c and Supplementary Fig. 3f. These results suggest that Ash1 complex is recruited to its targets by a factor recruiting Ash1, and Mrg15's presence within Ash1 complex may help to stabilize the chromatin association of Ash1 complex. We also rephrased our results part accordingly.

9. Is H3K36me2 deposition affected in Mrg15 KD? An heat map would also help in this case

Yes, H3K36me2 deposition was affected in Mrg15 knockdown at Ash1 peaks using heat map analysis (new Fig. 4a) or average analysis (new Fig. 4b).

10. Given the central role of H3K36me2 in transcription elongation, and the recent discovery of Mrg15 role in mRNA splicing (Luco & al Science, 2010, Gonzales et al, NSMB, 2014), the authors should provide a global view of mRNA expression level in Ash1 – Mrg15 KD cells. Ideally log2FC RNAseq (WT Vs KD) heat maps should be imposed over Ash1 heat maps centered on peaks (by imposing K-means clustering). Or, if difficult to visualize, a box plot measuring RNAseq log2FC (WT Vs KD) at genes right-on or close-to a given Ash1 peak, in both KDs.

We analyzed the RNA-seq results of all Ash1 target genes in WT, Ash1 KD and Mrg15 KD S2 cells. The result was incorporated into manuscript as Fig. 3e, 4e, 4f, Supplementary Fig. 3e. Our overall conclusion is that Ash1 affects the expression of only limited number of genes, and these genes are enriched in the set of Ash1 super target genes based on ChIP-seq intensity. The effect of Mrg15 knockdown at Ash1 target genes are relatively modest, this may be the way it is, or due to incomplete knockdown and/or the mixed roles of Mrg15 in other chromatin modifying complexes. Nevertheless, Mrg15-mediated Ash1 activation is physiologically relevant shown by genetics experiments at the organism level (Fig. 6).

Related to figure 5

The authors identified the MID domain of Ash1 and more specifically the Arg R1288 as a critical residue for Mrg15 interaction (Fig 5b, c). They also observed a reduced activity of Ash1-c R1288 as compare to Ash1-c.

11. Minor comment: Fig 5a – It would help the reader to present Ash1 truncations carton diagram ordered as loaded in Fig5b.

We appreciate the reviewer's careful reading and we have rearranged the carton diagram in Figure 5a as suggested.

Reviewer #2 (Remarks to the Author):

In this manuscript, Huang et al identified the novel Ash1 complex with Mrg15 and Nurf55 in *Drosophila* using S2 cells stably expressing Flag-tagged truncated Ash1C, and found that Mrg15 stimulates the catalytic activity of Ash1 in vitro. Further, in vivo, Mrg15 and Ash1 are recruited to the common target gene sites and Mrg15 is required for H3K36me2 around these sites. They also identified an interesting point mutant Ash1-R1288A, which abrogates its interaction with Mrg15, and its knock-in flies display multiple homeotic transformation phenotype, indicating the interaction of Ash1 with Mrg15 is critical for Ash1 function in fly. These findings are novel, highly original and informative and interesting to people working in not only TrxG-PcG fields but also general chromatin remodeling research. The methods they employed, such as Flag IP/mass spectrometry, ChIP-seq and data analysis by bioinformatics are straightforward, well standardized and well presented. Considering the points above, this reviewer suggests this manuscript is acceptable for the Journal. However, minor comment, albeit trivial, is as below.

Minor comment

On lines 69 and 150, the authors described “Mrg15 mediates allosteric activation of Ash1” or “via an allosteric activation mechanism”. They also presented kinetic data in Fig2. E, showing V_{max} is robustly activated albeit the K_m for nucleosomes are affected very small. This is in contrast to the concept of classical enzymology; hyperbolic saturation in Michaelis-Menten model vs sigmoidal allosteric model. Classical enzymology teaches us allosteric activation significantly increases the affinity for substrate, i.e. decreases the K_m , without much effect on the turnover V_{max} of enzyme. If authors stick to the term “allosteric”, they had better reconsider this comment.

The term “allosteric” is often used in two different scenarios. One is the classic allosteric activation of homo-multimeric enzyme complexes, that an initial binding event between enzyme and substrate allosterically changes the structure of the enzyme complex and facilitates subsequent binding events. Another scenario is that a regulatory subunit alters either substrate binding or turnover of the catalytic subunit, which is reflected by reduced K_m or increased V_{max} , respectively. We think that Mrg15-Ash1 complex belongs to the latter case of the second scenario. We agree with the reviewer that the term “allosteric” can be confusing due to abovementioned reasons, and we have removed the word “allosteric” in the manuscript as suggested.

Reviewer #3 (Remarks to the Author):

The molecular nature of the Ash1 complex and its biochemical activity have been slow to be characterized, after several false starts that resulted in incorrect claims. It is clear now that it is a histone H3K36 methyltransferase that acts in concert with Trithorax as an antagonist to Polycomb repression and promotes transcription. More complex is the question of the molecular nature of Ash1 and how it is recruited. The present paper makes

significant progress in characterizing the protein complex associated with Ash1. Affinity purification of FLAG-tagged truncated, full length Ash1 and human Ash1L, followed by mass spectrometry showed that the Ash1 complex includes Mrg15 and Nurf55 or the human orthologues of these proteins. The paper shows very competently that these components have no intrinsic methyltransferase activity but that Mrg15, with or without its chromodomain, greatly stimulates the catalytic activity of Ash1. This is an important advance that will be relevant to all who study Polycomb repression, its regulation, Trithorax, Ash1 and associated mechanisms in development and disease.

A few comments:

p.7 line 138. Figure 2A,B,C. It appears from the figure that Mrg15 delta chromodomain is substantially less potent at stimulating Ash1 catalytic activity. This impression from the autoradiography is in part corrected by the scintillation counts plotted below. It would be useful to clarify this point.

We appreciate that the reviewer carefully pointed out the difference. Given that some of the experiments shown in the last version were not done side-by-side and the exposure time can interfere, we performed additional experiments side-by-side during revision and we agree with the reviewer that Mrg15 mutant lacking chromodomain is less effective than the full length Mrg15. We also added a mutant Mrg15 lacking the MRG domain, which completely lost its stimulating ability. These results are included as new Supplementary Fig. 2d and 2e. To more accurately interpret the data, we added the following sentence into the results part: "These results suggest that the MRG domain of Mrg15 is the primary activator of Ash1, while the chromodomain of Mrg15 may participate in the stimulation process by stabilizing the structure of the Ash1-Mrg15 enzyme complex."

p. 8. Given its involvement in other complexes, it is not surprising that Mrg15 binds to many genomic sites in the absence of Ash1. At sites common to Ash1 and Mrg15, the authors show that Ash1 recruits Mrg15. However, more than half of the Ash1 binding peaks in the genome are unaccompanied by Mrg15. We don't know how Ash1 is recruited and although Ash1 can further recruit MRG15, it is clearly not sufficient: some other signal or component is required.

Reviewer 1 raised a similar question. During revision, we analyzed all the Ash1 peaks and realized that good amounts of Mrg15 were present at many Ash1 peaks that were not called as Mrg15 peaks in Fig. 2a. This is likely due to the arbitrary cut off during peak calling. Therefore, we performed more bioinformatics analysis. Our new observations are summarized as the following: 1. At all Mrg15 peaks, Ash1 is not apparently enriched, likely due to Mrg15's presence in multiple chromatin modifying complexes and Ash1 complex only contributes to a small fraction of Mrg15's chromatin occupancy (new Supplementary Fig. 3c); 2. At the majority of Ash1 peaks, Mrg15 is clearly enriched and such enrichment is Ash1-dependent (new Fig. 3b, 3c, Supplementary Fig. 3d); 3. In the previous version, we mentioned that Ash1 occupancy was slightly reduced upon Mrg15 knockdown, during revision, we realized that such effect was stronger at weak Ash1 peaks. Therefore we performed more analysis and included them as new Fig. 3c and Supplementary Fig. 3f.

These results suggest that Ash1 complex is recruited to its targets by a factor recruiting Ash1, and Mrg15's presence within Ash1 complex may help to stabilize the chromatin association of Ash1 complex. We also rephrased our results part accordingly. We agree that the recruiting factor for Ash1 complex remains unknown and we added a paragraph into the discussion regarding this important question.

Although the authors have compared the genomic distributions of Ash1, Mrg15 and transcriptional activity, they tell us very little about them. Figure 3B and Suppl. Figure 3C show the RNA tracks for only part of the region indicated in the figures. Are all genes that have Ash1 + Mrg15 transcriptionally active? Are genes that have Ash1 alone still active? We analyzed the RNA-seq results of all Ash1 target genes in WT, Ash1 KD and Mrg15 KD S2 cells. The result was incorporated into manuscript as Fig. 3e, 4e, 4f, Supplementary Fig. 3e and Supplementary Table 3. Our overall conclusion is that Ash1 affects the expression of only limited number of genes, and these genes are enriched in the set of Ash1 super target genes based on ChIP-seq intensity. The effect of Mrg15 knockdown at Ash1 target genes are relatively modest, this may be the way it is, or due to incomplete knockdown and/or the mixed roles of Mrg15 in other chromatin modifying complexes. Nevertheless, Mrg15-mediated Ash1 activation is physiologically relevant shown by genetics experiments at the organism level (Fig. 6).

p. 9 line 179. This is incorrect. The major H3K36 methyltransferase in *Drosophila* is Set2. There is a gene named NSD (Nuclear receptor binding SET domain protein), also known as Mes-4 or dMes-4 (after the *C. elegans* Mes-4) which also encodes a Set2-like methyltransferase but it is not the main H3K36 methylase.

Set2 is the major enzyme for H3K36me3, which also contributes to H3K36me2, especially at the gene bodies of active genes. Mes4 is an H3K36me2-specific enzyme, which was reported to be the major enzyme for H3K36me2 (ref. 79). We have rephrased the sentence as the following "Because Mes-4 and Set2, the other two H3K36 methyltransferases, are responsible for global H3K36 methylation in *Drosophila*".

p. 9 line 182. H3K36me2 is almost abolished upon depletion of Ash1. Since Ash1-binding genes are presumed to be transcriptionally active, this implies that Ash1 excludes the normal Set2 activity. How is this explained?

Set2-mediated H3K36me2 and H3K36me3 are transcription coupled. Upon Ash1 depletion, the transcription of certain target genes are abolished which will consequently lead to the relief of elongating Pol II engaged Set2, resulting in the removal of H3K36 methylation catalyzed by both Ash1 and Set2.

Suppl. Figure 3C. it is interesting to note that H3K36me2 virtually disappears over the Grip gene upon Ash1 knockdown although this gene normally binds little or no Ash1. Although it is possible that Grip expression depends on some unknown Ash1 target gene, it would be useful to know the transcription status of this gene, which is apparently not given in the figure.

We added the RNA-seq results of Grip gene into the figure (currently new Supplementary

Fig. 3e). Grip gene is actually a weak Ash1 target gene, whose expression is sensitive to Ash1 depletion, and is also slightly reduced upon Mrg15 depletion.

There have been claims that Ash1 affects transcription more globally than just at the sites of the major peaks. It would be very interesting to give some more attention to these results. Figure 3b and Suppl. Figure 3c show only the part of the data corresponding to common peaks of Ash1 and MRG15. More discussion of the relationship of Ash1 to transcription is needed.

We added the following paragraph into the discussion “Another interesting question is how broad does Ash1 complex regulate gene transcription. In the literature, this has been a debate (47, 76), our overall conclusion is that Ash1 affects the expression of only limited number of genes, and these genes are enriched in the set of Ash1 super target genes based on ChIP-seq intensity (Fig. 4d, 4e). This is in support to a previous report (76). In S2 cells, the effect of Mrg15 knockdown at Ash1 target genes was relatively modest, and future investigations would answer whether this was attributed to incomplete knockdown and/or the mixed roles of Mrg15 in other chromatin modifying complexes. Nevertheless, Mrg15-mediated Ash1 activation is a physiologically relevant event supported by genetic evidences at the organism level (Fig. 6).”

p. 11 line 212. This is an excellent approach.

Thanks and we appreciate!

p. 13, line 264. A sentence of explanation would be useful

Thanks for the suggestion and we rephrase the sentence as the following “As expected, overexpression of wild-type Ash1 fully rescued the haltere to wing transformation phenotype in *ash1^{22/R1288A}* mutant flies, whereas overexpression of Mrg15 and Nurf55 had little effect (Fig. 6d), consistent with the biochemical observations that Nurf55 could not stimulate Ash1 activity (Fig. 2a) and Mrg15 displayed greatly attenuated stimulation towards Ash1C-R1288A (Fig. 5d).”

The discussion is a brave attempt but disappointing. The paper makes important progress in giving us a clear idea of the components of the Ash1 complex and their effects on catalytic activity but the key questions remain unanswered and mysterious.

How is Ash1 recruited? How does it antagonize Polycomb repression? Is it really through H3K36me2 inhibition of PRC2? If so, why does the H3K36 methylation produced by Set2 not sufficient? How is Ash1 related to Trx? These are some of the key unanswered questions. The discussion has little to say about them.

We agree that many important questions remain to be studied as the reviewer raised, for example, the recruitment mechanism of Ash1 complex, the way how Ash1 interplays with other TrxG factors, etc., which we added into discussions as important issues for further exploration.

Regarding the reviewer's question about whether Ash1 antagonizes Polycomb repression through H3K36me2 inhibition of PRC2. We think this is likely the case for the following reasons: (1) We know that a group has generated an *ash1* knock-in *Drosophila* line with a

point mutation at the catalytic site, and this mutant shows homeotic transformation as severe as *ash1* null allele, which strongly demonstrates that H3K36me2 catalysis is vital for Ash1's polycomb-antagonizing function (unpublished personal communication); (2) Our study also shows that impairment of the full HMT activity of Ash1 complex by attenuating the binding and stimulation of Mrg15 towards Ash1 compromises the anti-polycomb function of Ash1; (3) We do believe that Set2-mediated H3K36me3 methylation is also capable of antagonizing PRC2 activity (although maybe not at the Hox genes, because Set2 may not be as abundant at the Hox genes), because among all histone methylations examined at the genome-wide level, H3K36me3 is actually the best one that negatively correlates with H3K27me3 (1-6).

It could be abbreviated and let the data speak for themselves.

According to the reviewer's suggestion, we removed certain paragraphs unrelated to the central role of Ash1 that is anti-Polycomb silencing.

References:

1. Mikkelsen TS, Ku M, Jaffe DB, Issac B, Lieberman E, Giannoukos G, et al. Genome-wide maps of chromatin state in pluripotent and lineage-committed cells. *Nature*. 2007;448(7153):553-60.
2. Young MD, Willson TA, Wakefield MJ, Trounson E, Hilton DJ, Blewitt ME, et al. ChIP-seq analysis reveals distinct H3K27me3 profiles that correlate with transcriptional activity. *Nucleic Acids Res*. 2011;39(17):7415-27.
3. Schmitges FW, Prusty AB, Faty M, Stutzer A, Lingaraju GM, Aiwazian J, et al. Histone methylation by PRC2 is inhibited by active chromatin marks. *Mol Cell*. 2011;42(3):330-41.
4. Yuan W, Xu M, Huang C, Liu N, Chen S, Zhu B. H3K36 methylation antagonizes PRC2-mediated H3K27 methylation. *J Biol Chem*. 2011;286(10):7983-9.
5. Ferrari KJ, Scelfo A, Jammula S, Cuomo A, Barozzi I, Stutzer A, et al. Polycomb-dependent H3K27me1 and H3K27me2 regulate active transcription and enhancer fidelity. *Mol Cell*. 2014;53(1):49-62.
6. van Galen P, Viny AD, Ram O, Ryan RJ, Cotton MJ, Donohue L, et al. A Multiplexed System for Quantitative Comparisons of Chromatin Landscapes. *Mol Cell*. 2016;61(1):170-80.

REVIEWERS' COMMENTS:

Reviewer #1 (Remarks to the Author):

ready for publication

Reviewer #2 (Remarks to the Author):

Minor concern of this reviewer was fully answered.

Reviewer #3 (Remarks to the Author):

The authors have clarified several points, corrected others, and modified several figures. I think the manuscript is substantially improved and is now appropriate for Nature Communications.